# Development and Validation of Scientific Inquiry Literacy Instrument (SILI) Using Rasch Measurement Model

Dina Rahmi Darman [1,2,*], Andi Suhandi [1,*], Ida Kaniawati [1], Achmad Samsudin [1] and Firmanul Catur Wibowo [3]

1    Department of Science Education, Universitas Pendidikan Indonesia, Bandung 40154, Indonesia; kaniawati@upi.edu (I.K.); achmadsamsudin@upi.edu (A.S.)
2    Department of Primary Teacher Education, Universitas Negeri Jakarta, Jakarta 13220, Indonesia
3    Department of Physics Education, Universitas Negeri Jakarta, Jakarta 13220, Indonesia; fcwibowo@unj.ac.id
*    Correspondence: dina_rd@upi.edu (D.R.D.); andi_sh@upi.edu (A.S.)

**Abstract:** This research aims to develop an instrument of knowledge and skills called the Scientific Inquiry Literacy Instrument (SILI). Instrument measurement analysis uses the Rasch model by utilizing the Winsteps application. This research uses mixed methods research that combines interviews as qualitative research and tests as quantitative research. The research design used is Sequential Exploratory. The SILI contains 28 knowledge aspect indicators and 37 skills aspect indicators. Based on the Outfit mean square (MNSQ), Z-Standard value (ZSTD), and point measure correlation value (Pt Mean Corr), it was obtained that the three aspects of the SILI meet the criteria for an acceptable item in the Rasch measurement model. Based on item analysis biased towards gender, region, and science major, all SILI items in knowledge and skills meet the probability criteria > 0.05, so all items can be used without harming one group. The findings of the research dimensionality of the SILI for aspects of knowledge and skills are 26.9% and 20.4%. Thus, all aspects of the SILI can measure what should be measured and fulfill all the criteria for items accepted based on the Rasch model for measuring science inquiry literacy.

**Keywords:** development and validation; scientific inquiry literacy instrument; Rasch model

## 1. Introduction

Science education programs are required to train actual scientists by paying attention to skills and competencies [1]. The learning of prospective science teachers must be distinct from the nature of science (NoS) [2]. Science is seen as a body of knowledge, a way of thinking and investigating. Science for prospective teacher students is a collection of knowledge from observations and research that explain what, why, and how a phenomenon occurs [3,4].

Many science teachers need to understand the nature of science, and teaching science does not use the NoS [5]. So, the NoS was reimagined in various ways, including using social media [6]. Then, for prospective science teachers, inquiry-based learning activities are provided to increase the understanding of the NoS [7]. It turns out that implementing science education that uses the NoS and inquiry is not easy; many obstacles are found, such as the coherence of the learning curriculum, content, and support for its implementation [8].

Many students need help understanding the nature of science. So, much research was carried out to improve students' views about the NoS, one of which was through scientific investigation [9]. Other research was conducted to overcome the limitations of students' conceptions of NoS by providing a history of science, scientific explanations, and socio-scientific issues [10]. Then, a collaborative follow-up investigation was carried out on high school students to provide an understanding of the NoS and inquiry [11]. In addition, explicit NoS teaching and argumentation positively impact science learning [12]. Teaching

science by familiarizing students with conceptualization can facilitate the introduction and habituation of the NoS [13].

Prospective science teachers' science learning involves using several of the five senses, hands-on and minds-on, involving inquiry activities [14,15]. Inquiry activities carried out by scientists are carried out to discover concepts, principles, laws, or theories [16]. Inquiry activities have been adopted as a method in science-learning activities oriented to learning outcomes, including the knowledge, skills, and attitude domains [17,18]. Prospective teacher students are trained to teach science with inquiry so that science learning is by the NoS and produces students accustomed to conducting inquiry when solving problems in everyday life [19]. Prospective science teachers must understand the NoS with good scientific inquiry literacy (SIL) as an essential provision for inquiry teaching.

Scientific inquiry literacy (SIL) bridges scientific literacy and inquiry [20]. Scientific literacy can be defined as scientific knowledge and skills consisting of process skills and scientific understanding [6,21]. Scientific literacy is the ability to apply and master science to solve problems in life [22]. Scientific literacy is acquiring knowledge, scientific reasoning, and critical thinking to build scientific knowledge [23]. Meanwhile, scientific inquiry literacy is a part of scientific literacy that cannot be separated from and is interconnected with one another [24]. Scientific inquiry leads to various ways scientists study the world and produce explanations based on the evidence they obtain [25]. Scientific inquiry also refers to developing a knowledge and understanding of scientific ideas and how scientists study nature [26]. A person who carries out scientific investigation activities acquires scientific knowledge and skills to carry out scientific investigations. SIL is a person's ability to understand and apply scientific concepts in scientific investigations using scientific steps [27,28]. SIL can also be interpreted as scientific skills and knowledge, consisting of process skills and scientific understanding [29,30].

The science development process closely relates to SIL by applying knowledge and utilizing science [31]. SIL is also closely associated with the ability to process information and conduct scientific inquiry activities [32]. SIL bridges scientific literacy and scientific inquiry [20]. SIL not only requires understanding science content but also acquiring and developing science process skills, creativity, and critical thinking to develop scientific knowledge and inquiry activities carried out by scientists [33]. An essential component in scientific inquiry literacy is the NoS in the form of scientific processes, content, and attitudes [20,34]. Based on the description above, SIL is defined as the mastery of aspects of scientific inquiry, namely the ability to understand and apply scientific concepts in the inquiry process, which includes content, processes, and attitudes. So, scientific inquiry literacy is very important for science teachers, and knowledge, skills, and attitude are part of the learning outcomes in inquiry activities for prospective science teachers. If science teachers do not have good SIL, then students will not receive science learning by the NoS.

The Scientific Inquiry Literacy Instrument (SILI) can accurately measure the level of scientific inquiry literacy of prospective teacher students. Wenning developed an instrument called ScInqLiT for SIL [27]. ScInqLiT was developed based on nine stages of scientific inquiry. ScInqLiT was created as a paper-based test. In this test, students are asked to design a science experiment, draw conclusions based on experiments, interpret numerical data, and conclude the experiment based on numerical data. Then, Innatesari developed eight SILI indicators based on ScInqLiT and the science inquiry stage. The eight SILI indicators were formed into 24 multiple-choice items through written tests adapted to the school curriculum [35]. Then, Kusnadi developed eight SILI indicators based on inquiry skills and ScInqLiT, including scientific inquiry, principles, and the implementation of project-based inquiry [36]. Another instrument designed to measure scientific inquiry is the Views of Scientific Inquiry (VOSI), which consists of five questions about aspects of scientific inquiry [37]. However, because there was an incomplete part of the scientific inquiry, it was refined into the Views About Scientific Inquiry (VASI) [38]. The VASI covers eight aspects of scientific inquiry [39], using 24 open-ended questions using pencil and paper [40].

Based on the analysis of the SILI that has been developed, each instrument has its advantages, such as ScInqLiT items, which contain scientific inquiry skills. The VASI items include scientific inquiry knowledge. The instrument analysis found that no SILI consists of two parts of the scientific inquiry learning outcomes: knowledge and skills. So, developing a SILI that includes the three parts mentioned above is necessary to evaluate prospective teacher students' SILI achievements.

Instrument development is an activity that creates measuring tools in education that are used for research [28,41]. An instrument is an educational measuring tool that meets the requirements for measuring an object or collecting data about a variable [42]. Instrument validation is a method used by researchers to measure the validity of an instrument [43,44]. Validity is the degree of accuracy between the data on the research object and the power that can be reported by the researcher [45,46]. Valid data match the reported data with the actual data on the research object [47]. A measurement instrument in education is valid if it can measure what it is intended to count or measure accurately [48]. The purpose and subject matter determine whether an instrument is valid [49]. Determining the validity of an instrument depends on considerations for what and for whom it is used [50]. In broad terms, the validity of an instrument depends on the specific objectives and subjects of measurement. Mixed methods in science education research are used to develop instruments [51] and interdisciplinary research [1]. This research consists of quantitative and qualitative research [52]. A mixed methods approach can be used to determine prospective teachers' perspectives in higher education [53].

The Rasch model is a modern theory of item assessment [54,55]. The Rasch model in instrument development responds to various weaknesses of the classical test theory (CTT) paradigm [56,57]. The fundamental difference between the two theories lies in the raw score analysis process [56]. In CTT, raw score analysis is carried out directly in a rating scale and treated as data that look like an integer. However, raw data cannot be then analyzed using the Rasch Model. Instead, they must first be converted into an 'odds ratio' and then logarithmically transformed into logit units to determine the respondent's probability of responding to an item. The Rasch model can restore data according to its natural conditions. The natural conditions refer to the fundamental nature of quantitative data, namely that they are continuous [58]. CTT, which uses raw data from rating responses, cannot present the original characteristics of continuous quantitative data. Through the Rasch model, ordinal responses can be converted into ratios with a higher accuracy level by referring to the principle of probability [55].

The Rasch model has the advantage of classifying the calculation of question items and people in the person-item map [59,60]. There are two principles underlying the Rasch model [61]. The first is the subject's ability to answer questions that can be predicted using characteristics. The second principle describes the relationship between the subject's ability and the question item, or the relationship between the question and another ability, which is depicted as an item characteristic curve [59]. Compared to other methods, especially classical test theory, the advantage of the Rasch model is its ability to predict test takers' probabilities correctly. Test takers with high ability should have a greater chance of answering questions correctly than other students and vice versa. The next advantage is that the Rasch model does not only pay attention to items but also to aspects of the response and the correlation between items and responses [61].

Furthermore, Rasch modeling is superior in predicting missing data based on systematic response patterns [62,63]. Because of these advantages, it became the basis for developing a SILI using the Rasch model with Winsteps 5.4.1. Based on the explanation above regarding the importance of SIL and its measurement instruments, which cover all aspects of scientific inquiry learning outcomes that are by the nature of science, it is necessary to develop SIL instruments that improve previous instruments. The development of the instruments above will provide results that can be a reference for science teachers and other scientific communities using Rasch model measurements. So, research is needed to produce a SILI consisting of aspects of knowledge and skills using the Rasch model, which

is the aim of this research. The basis for the investigation is in the form of the following research questions.

Q1: What are the aspects of evaluating the SILI of prospective science teachers?

Q2: What are the SILI indicators in the knowledge aspect to evaluate the SILI of prospective science teachers?

Q3: What are the SILI indicators in the skills aspect that can be used to evaluate the SILI of prospective science teachers?

Q4: How do prospective science teachers respond to the two aspects of the SILI?

## 2. Materials and Methods

### 2.1. Research Method

This research aims to design an instrument and develop SILI aspects of knowledge and skills, which are analyzed using the Rasch model and the Winsteps application. This research design uses a mixed method study. The design used is Sequential Exploratory. This research combines interviews as qualitative research and tests as quantitative research. The Sequential Exploratory research procedure on the development of the SILI is presented in Figure 1.

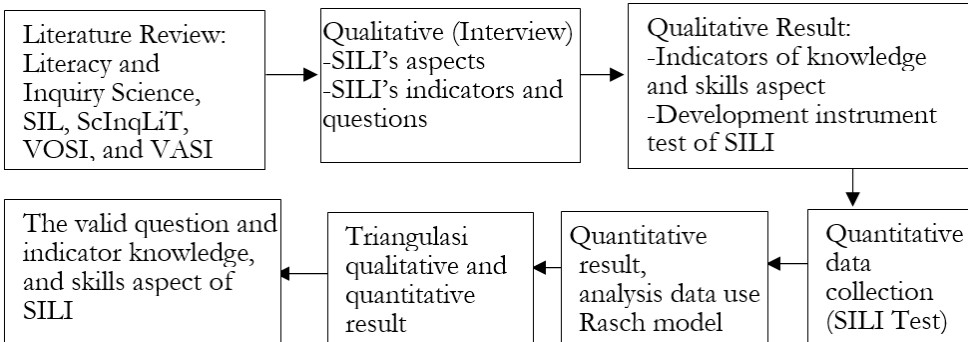

**Figure 1.** The Sequential Exploratory research procedure on the development of the SILI.

The research began with a literature study on SIL, scientific literacy, scientific inquiry, and instruments that had been developed previously, covering stages of inquiry, aspects of scientific literacy, and indicators. Then, SILI aspects and indicators were designed to be used as a basis for qualitative research. Five experts validated SILI aspects, indicators, and items using interviews at the qualitative research stage. This stage seeks input and suggestions from experts for the development of the SILI. Based on the results of interviews with experts, SILI improvements were carried out. This stage was repeated five times until experts declared the SILI validated and could be continued for quantitative studies. Next, quantitative research was carried out using the SILI trial test that was developed. Data from the test results were analyzed using the Rasch measurement model. The next stage is interpreting quantitative data by considering suggestions and input from experts. Based on the interpretation results, the SILI meets the valid criteria and can be used to measure SIL in knowledge and skills.

### 2.2. Research Context

This research involves universities producing prospective science teachers in Indonesia. The reason for choosing the SILI is that it is an essential ability for science teachers, so only science students can participate. The universities chosen are only state universities familiar with inquiry in science learning. Universities that do not provide inquiry skills in the teaching of prospective science teachers are not considered. The second aim of this research is to evaluate prospective science teachers' SILI. The third aim is easy access to all prospective science teachers so that only science teacher responses are taken that are easy to obtain. Indonesia's science teacher education program consists of four science education study programs: physics, chemistry, biology, and science. The physics education study

program aims to prepare prospective physics science teachers at the high school level; the chemistry education study program aims to prepare prospective chemistry science teachers at the high school level; the biology education study program aims to prepare prospective biology science teachers at the high school level, and science education aims to prepare prospective science teachers at the junior high school level. Prospective science teachers who are ready to teach science at the high school level, namely physics, chemistry, and biology science teachers, are allowed to teach science to middle school and elementary school students. Students who graduate from the science education study program are qualified to teach at a lower level of education, namely elementary school. This condition is adjusted to the needs and conditions in the field. So, all prospective science teachers need to have a good SILI, and an evaluation of their SILI abilities needs to be carried out to be ready to teach scientific inquiry well.

*2.3. Sample Research*

The sample for qualitative research consists of five science experts regarding SILI aspects, indicators, and items using interviews. The experts include two science education professors and three Doctors of Science Education. The sample for quantitative research consists of 201 participants. Table 1 displays demographic information about the participants involved in developing the SILI.

**Table 1.** Demographic profile of SILI development participants.

| Profile Demography | Category | Frequently | Percentage |
|---|---|---|---|
| Gender | Male | 36 | 17.91 |
| | Female | 165 | 82.09 |
| Year at University | 1 | 67 | 33.33 |
| | 2 | 53 | 26.36 |
| | 3 | 66 | 32.83 |
| | 4 | 11 | 5.47 |
| | >4 | 4 | 1.99 |
| Region | West Java | 97 | 48.25 |
| | Sister City | 48 | 23.88 |
| | Central Java | 15 | 7.46 |
| | Sumatera | 26 | 12.93 |
| | West Java | 10 | 4.97 |
| | Other | 5 | 2.28 |
| Major at University | Physics Education | 47 | 23.38 |
| | Chemistry Education | 25 | 12.43 |
| | Biology Education | 36 | 17.91 |
| | Science Education | 93 | 46.26 |

Participation in this study was carried out voluntarily without any coercion. All personal data collected are kept confidential and not shared. Personal data only facilitate coding for participant data collection and analysis. All participants are prospective teachers from physics, chemistry, biology, and science groups. Differences in scientific background are not considered a determining factor. Based on Table 1 the male participants comprised 36 people and the female participants included 165. Participants in this study came from five regions, namely West Java, Sister City, Central Java, Sumatra, West Java, and others. Participating students consisted of students in year 1 (67 students), year 2 (53 students), year 3 (66 students), year 4 (11 students), and students who were still waiting for graduation

in year > 4 (4 students). Apart from that, they came from different major science groups, namely physics (47 students), chemistry (25 students), biology (36 students), and science (93 students).

### 2.4. Instrument Research

The instrument used in the qualitative research is an interview sheet, and the instrument used in the quantitative research is the SILI test, which is the result of qualitative research. The SILI comprises 35 items regarding knowledge and 39 skills aspect questionnaires designed in this research. The instrument was developed based on aspects of inquiry [27], the NoS [38], and scientific literacy [64]. The knowledge aspect consists of 28 indicators in Table 2.

**Table 2.** SILI indicators for knowledge aspects.

| Component | SILI Indicators for Knowledge Aspect | SILI's Items |
|---|---|---|
| 1. Knowledge of definitions, understandings, terms, types, and inquiry positions | Knowledge of the meaning of inquiry | 1 |
| | Know the wisdom that underlies the inquiry approach | 6 |
| | Know other terms for inquiry | 2, 4 |
| | Know the types of inquiry in learning | 33 |
| | Know the position of inquiry in learning | 34 |
| 2. Knowledge of concepts related to inquiry | Knowledge of the concept of observation in inquiry | 3 |
| | Distinguish between practice, observation, experiment, measurement, and practicum | 9 |
| | Know the experimental activities of inquiry well | 8 |
| | Choose the right concept from the image related to the inquiry | 14 |
| | Know the concept of measurement in inquiry | 15, 31 |
| | Be well acquainted with the concept of generalization | 19 |
| | Get to know the concept of the scientific method well | 20 |
| | Get to know the parts of the scientific method well | 35 |
| | Familiar with the concept and orientation of inquiry experimental activities | 21, 32 |
| | Get to know the concept of classification well | 22, 26 |
| 3. Knowledge of supporting skills for inquiry activities | Good knowledge of science process skills | 27, 29 |
| | Know observation skills | 23 |
| | Know classification skills | 22 |
| | Know about measurement skills | 25 |
| | Know about communication skills | 18,24 |
| | Get to know about generalization (concluding) skills | 30 |
| | Be well acquainted with the concept of predictive ability | 28 |
| 4. Knowledge of the steps for inquiry activities | Knowledge of the concept of hypothesis | 5 |
| | Know the form of a guide for scientific inquiry activities | 7 |
| | Get to know the tools and materials for science experimental activities | 10, 11 |
| | Get to know the types of variables in science practical activities | 12, 13 |
| | Know the presentation of data from inquiry practicum results | 16 |
| | Know the differences between graphs and diagrams as a result of inquiry activities | 17 |

Based on Table 2, the SILI for the knowledge aspect consists of multiple-choice questions with five answer choices. The SILI knowledge aspect contains prospective teacher students' knowledge of SILI concepts that support SIL. These concepts consist of the concepts of inquiry, observation, hypothesis, foundations of the inquiry approach, guidelines for inquiry activities, terms experiment, experiments, practicum and practice, measurement variables, scientific communication, graphs and diagrams, generalizations, scientific methods, the concepts and orientation of inquiry experimental activities, classification, process science skills, and predictive abilities. The SILI skills aspect consists of stages of scientific inquiry, outlined in the indicators presented in Table 3, which detail each question's indicators of the SILI skills aspect.

**Table 3.** SILI indicators of skills aspects.

| No. | Stages of Scientific Inquiry | Indicators of SILI Questions for Skills Aspects | SILI's Items |
|---|---|---|---|
| 1 | Identify that problem will be investigated | Identify that the problem will be investigated from the given phenomenon | 29 |
| 2 | Use of deduction, formulation of hypotheses, or combining logical models and proofs | 2.1. Formulate a best hypothesis from a scientific problem | 12, 38 |
| | | 2.2. Formulate and revise scientific explanations and models with the use of logic and evidence | 20 |
| | | 2.3. Give scientific proof to support the claim | 22 |
| | | 2.4. Provide an evaluation of the given hypothesis | 24 |
| | | 2.5. Provide an explanation hypothesis based on the condition's beginning and end phenomenon | 30 |
| 3 | Use deduction to produce predictions from hypotheses or models | 3. Use deduction from a law to make predictions | 37 |
| 4 | Design experimental procedure for test predictions | 4. Sequence the experimental science process presented in a random way | 1 |
| 5 | Conduct scientific experiment, observation, or simulation to test hypothesis or model: 1. Identify test system; 2. Identify and define variables in operational way; 3. Perform experiment | 5.1. Refine the test design with a certain objective | 4 |
| | | 5.2. Design a test using lots of variables | 9 |
| | | 5.3. Determine the best way to collect data for a scientific investigation | 33, 34 |
| | | 5.4. Explain the error inside a test | 5 |
| | | 5.5. Select and explain the suitable material for the test based on the list of ingredients in the table | 25 |
| | | 5.6. Explain error variables in the test | 3 |
| | | 5.7. Determine and provide an accurate reason for taking samples in data collection in an experiment | 26 |
| 6 | Collect, organize, and analyze data thoroughly, accurately, and precisely: 1. Analyze data for trends and relationships; 2. Create and interpret chart; 3. Use induction and develop law based on evidence and graph | 6.1. Interpret data based study results | 14 |
| | | 6.2. Determine valid data based on the picture distribution of data presented | 27 |
| | | 6.3. Determine a valid way of data collection based on the situation | 28 |
| | | 6.4. Explain the data deviation in a test results chart | 15 |
| | | 6.5. Make an interesting conclusion based on the data in the test results graph | 16 |
| | | 6.6. Determine the most appropriate variable based on the data in the graph | 18 |
| | | 6.7. Give the meaning of the trend based on the data presented in the form of data table | 23 |
| | | 6.8. Explain the meaning of data given in a graph that intersects at an axis | 19 |

**Table 3.** *Cont.*

| No. | Stages of Scientific Inquiry | Indicators of SILI Questions for Skills Aspects | SILI's Items |
|---|---|---|---|
| 6 | Collect, organize, and analyze data thoroughly, accurately, and precisely:<br>1. Analyze data for trends and relationships;<br>2. Create and interpret chart;<br>3. Use induction and develop law based on evidence and graph | 6.9. Make an interesting conclusion based on the data in the table data provided | 36 |
| | | 6.10. Provide an opinion to state something based on the data in the graph | 6 |
| | | 6.11. Interpret the given graph based on the results of the data observation | 10 |
| | | 6.12. Make an interesting conclusion based on the analysis of the data in the graphs presented | 17 |
| | | 6.13. Create a chart based on the given data | 39 |
| | | 6.14. Create a decision based on experimental data | 13 |
| 7 | Apply numerical and statistical methods to obtain and support conclusion:<br>1. Use technology and mathematics;<br>2. Make interesting, correct conclusion from proof | 7.1 Provide the correct reason for data processing a test to obtain a conclusion | 32 |
| | | 7.2. Give an opinion for correct conclusions drawn from an experiment (an interesting conclusion from evidence) | 2 |
| | | 7.3. Make an interesting conclusion based on a graph of the given data | 11 |
| | | 7.5. Make an interesting conclusion comparing quantitative/qualitative ways or subjective/objective data | 31 |
| | | 7.6. Declare agreement or no agreement, along with the reason for the conclusions presented based on the proof | 35 |
| 8 | Explain unexpected results:<br>1. Formulate hypothesis or alternative models;<br>2. Identify and communicate sources; errors cannot be avoided;<br>3. Identify reason for no consistent results | 8.1. Recognize and analyze alternative explanations and models | 21 |
| | | 8.2. Give a reason for data that were obtained that were not reasonable in a graph (experiment error) | 8 |
| | | 8.3. Give a reason for a found trend or relationship that is not fair in the graph | 7 |

### 2.5. Data Collection

Data collection in quantitative research uses a validation sheet based on interviews. The sheet consists of aspects, indicators, and statements about the SILI. Science experts responded and provided comments and suggestions on indicators in all aspects. They also checked the suitability of the questions and indicators in the SILI. This process was carried out three times until appropriate indicators and questions were found for the SILI.

Data collection in quantitative research uses the SILI test and consists of four parts. The first part consists of the respondent's consent to participate in the study; the second part is the respondent's identity, which contains personal data, and its confidentiality is guaranteed. The third part consists of the SILI knowledge aspect comprising 36 items. This section is a closed question with five answer choices chosen by respondents according to their scientific inquiry knowledge. To complete this aspect, the questionnaire is estimated to take 30–35 min. The fourth section consists of the SILI skills aspect, which consists of 39 items. This section consists of questions with four answer choices determined by respondents according to their scientific inquiry skills. To complete the skills aspect of the SILI is estimated to take 35–40 min. The validation was carried out according to the Rasch model to see the construct validation using Winsteps version 5.4.1. The SILI data collection procedures were implemented from October 2022 to June 2023.

### 2.6. Analysis and Interpretation

The quantitative data analysis was carried out by collecting data regarding SIL indicators at the interview stage with experts, eliminating unnecessary data, and categorizing SIL indicator data into two aspects, namely knowledge and skills. Then, the data were displayed, explaining the indicators and aspects of SIL as presented in Tables 2 and 3. Then, after obtaining valid indicators in the two aspects, SIL questions were prepared, which would be used to collect quantitative data. At the quantitative research stage, the data analysis was obtained from the results of test questions for prospective science teachers

regarding the SILI in the aspects of knowledge and skills. The results of student answers on the knowledge and skills aspect test are given a value of 1 for the correct answer and 0 for each wrong answer.

The validation of the SILI based on the Rasch model measurement was determined, as well as the suitability/mismatch of items and respondents, dimensionality, and differential item functioning analysis of items. These statistics indicate that the instruments and scales created for this research were appropriate for their purposes [48,65,66]. There are essential parts in an analysis using the Rasch model, namely Rasch statistics, person and item reliability, dimensionality, differential item functioning (DIF), targeting, the ability to calibrate and estimate items, the item characteristic curve in the parameter models, the information function of items and instruments, and the interaction map between items and respondents [54,55,67].

### 2.6.1. Rasch Statistics

Rasch statistics consist of the Outfit mean square (MNSQ), Outfit Z-Standard value (ZSTD), Point measure correlation value (Pt Mean Corr), and Joint Maximum Likelihood Estimation (JMLE) Measure. These are obtained to explore the compatibility of the data with the model [68]. The difficulty level of each item is determined based on the JMLE Measure value. The greater the JMLE Measure value, the greater the item's difficulty level. The greater the JMLE Measure value, the higher the difficulty level and vice versa. Item reduction is carried out if the SILI item does not meet the abovementioned statistics. Criteria used to check the suitability of inappropriate question items (outliers or misfits) [66]:

a.    MNSQ value accepted: $0.5 < MNSQ < 1.5$.
b.    ZSTD received: $-2.0 < ZSTD + 2.0$.
c.    Pt Mean Corr received: $0.4 < Pt\ Measure\ Corr < 0.85$.

### 2.6.2. Person and Item Reliability

The Rasch model measurement can also determine reliability in the form of Cronbach's alpha, item reliability, and person reliability in a measure. The Cronbach alpha value shows the overall interaction between persons and items. KR20 has very good criteria if the value > 0.8 and good criteria if the value is 0.8–0.9. In addition, the Rasch model also provides information and separation functions for both items and persons. The item information function also shows the reliability of the measurements [69]. The dimensionality of an instrument is essential to evaluate whether the instrument being developed can measure what it is supposed to measure [70].

### 2.6.3. Dimensionality

The dimensionality referred to here is evaluating the extent to which the developed SILI can measure the SILI abilities of prospective science teachers [66]. According to the Rasch model, the minimum dimensionality requirement is 20%; if the value is more than 40%, it means it is better, and if it is more than 60%, it is special. Another aspect of measuring dimensionality is the variance that the instrument cannot explain (unexplained variance), ideally not exceeding 15% [59].

### 2.6.4. Differential Item Functioning (DIF)

Differential item functioning (DIF) is a Rasch Model analysis used to determine whether an item can be used or not to measure latent constructs in different groups of science teacher candidates but in the same way [68]. DIF provides information on differences between groups of prospective science teachers in SILI questions' difficulty levels. DIF analysis in this study was used to evaluate SILI items for gender, region, and science major. The DIF testing criteria are that if the question's probability value is more excellent than 5%, the question item is acceptable and does not harm certain groups.

### 2.6.5. Targeting

Person-item maps examine targeting in Rasch analysis [68]. The research target is to determine the differences between the average ability of prospective science teachers and the moderate difficulty of LIS items. The closer that the prospective science teacher's ability and the LIS item difficulty are means the better the expected target. On Wright maps, targeting scales and accepted differences are less than one logit. This condition shows that there was no error in targeting.

### 2.6.6. Wright Maps

Wright maps are one of the most valuable aspects of Rasch measurements, allowing researchers to quickly explain research results to readers [71]. Wright maps depict the distribution of prospective science teachers' abilities and the difficulty level of questions on the SILI. Wright maps present the person and item distribution of scores, the mean (M) of the person and item, and one (S) and two (T) standard deviations from the mean.

## 3. Results and Discussion

Data from testing questions on prospective science teachers regarding the SILI in knowledge and skills were analyzed using Winsteps software version 5.4.1 to confirm the SILI items developed and the respondents' SILI abilities. Processing the results of the responses to the SILI in the three aspects using Winsteps, the values obtained for Cronbach alpha coefficients (KR20), Outfit MNSQ, Outfit ZSTD, and separation are presented in Table 4.

**Table 4.** Cronbach alpha, reliability, MNSQ, and separation of SILI.

| SILI Aspects | KR20 | Person Measure | Reliability | | Infit MNSQ | | Outfit MNSQ | | Separation | |
|---|---|---|---|---|---|---|---|---|---|---|
| | | | Person | Item | Person | Item | Person | Item | Person | Item |
| Knowledge | 0.72 | 0.19 | 0.73 | 0.98 | 1.01 | 0.99 | 1.06 | 1.06 | 1.66 | 6.60 |
| Skills | 0.76 | −0.38 | 0.74 | 0.96 | 1.00 | 1.00 | 1.00 | 1.00 | 1.69 | 4.77 |

Table 4 shows that the reliability of SILI items for the knowledge aspect is 0.98 with a very reliable category, and item separation is 6.60. The value of the person measure of the SILI knowledge aspect is 0.19. This value shows the average response value of prospective science teachers in working on SILI items. An average value more significant than the logit value of 0.0 indicates a tendency for the ability of prospective science teachers to be greater than the level of difficulty of the questions. The Cronbach alpha value of the SILI knowledge aspect is 0.72, with good interpretation. The value of the person's reliability in the SILI knowledge aspect obtained is 0.73. So, it is interpreted that the consistency of the responses is in the medium category. The SILI item reliability value for the knowledge aspect is 0.98 with a special interpretation. So, the quality of the items on the SILI for the special knowledge aspect is nearly perfect. Meanwhile, the person reliability is 0.73 with sufficient criteria; the separation of the person is 1.66; the item reliability is 0.96, with a very reliable category and item separation of 4.77.

Meanwhile, the person reliability of the SILI skills aspect is 0.74 with sufficient criteria and a separation of 1.69; the value of the SILI person measure is −0.38, showing that the average response value of prospective science teachers is smaller than logit 0.0. This condition indicates a tendency for the ability of prospective science teachers to be lesser than the difficulty level of the questions. The Cronbach alpha value of the SILI skills aspect is 0.76, with good interpretation. The value of the person reliability instrument for the SILI skills aspect was 0.74. So, it is interpreted that the consistency of the responses is in the medium category. The SILI item reliability value for the skills aspect is 0.96 with a special interpretation. So, the quality of the items on the SILI for the special skills aspect is close to perfect. The results of the detailed analysis of respondent responses

provide information about the item characteristics, instrument dimensionality, item bias, and response characteristics of each aspect of the SILI.

### 3.1. Item Characteristics

3.1.1. Item Characteristics of the SILI Knowledge Aspect

The item characteristics of the SILI knowledge aspect are presented in Table 5. The SILI knowledge aspects items are sorted by difficulty level from difficult to easy questions based on the JMLE Measure. The higher the logit value, the more difficult the question item. The most difficult questions are S16, S18, and S32, while the easiest are S15, S14, and S5.

**Table 5.** Item characteristics of SILI knowledge aspect.

| Item | Total Score | JMLE Measure | Model S.E. | Infit | | Outfit | | Pt Measure-Al | | Exact Obs% | Match Exp% |
|------|------|------|------|------|------|------|------|------|------|------|------|
| | | | | MNSQ | ZSTD | MNSQ | ZSTD | Corr | Exp | | |
| S16 | 37 | 1.85 | 0.19 | 0.89 | −0.98 | 0.85 | −0.75 | 0.37 | 0.24 | 82.6 | 81.7 |
| S18 | 42 | 1.68 | 0.18 | 1.07 | 0.73 | 2.01 | 4.66 | 0.12 | 0.25 | 78.6 | 79.4 |
| S32 | 43 | 1.65 | 0.18 | 0.96 | −0.36 | 0.93 | −0.40 | 0.30 | 0.26 | 81.1 | 79.0 |
| S34 | 46 | 1.56 | 0.18 | 0.98 | −0.24 | 0.91 | −0.54 | 0.30 | 0.26 | 79.6 | 77.7 |
| S35 | 54 | 1.32 | 0.17 | 1.19 | 2.34 | 2.03 | 5.93 | −0.02 | 0.28 | 71.1 | 74.3 |
| S26 | 62 | 1.11 | 0.16 | 1.21 | 3.04 | 1.32 | 2.52 | 0.02 | 0.29 | 66.7 | 71.1 |
| S11 | 63 | 1.09 | 0.16 | 1.02 | 0.25 | 1.07 | 0.59 | 0.26 | 0.29 | 71.1 | 70.7 |
| S29 | 70 | 0.91 | 0.16 | 1.03 | 0.54 | 1.06 | 0.59 | 0.26 | 0.30 | 68.7 | 68.2 |
| S19 | 71 | 0.89 | 0.16 | 1.00 | 0.06 | 1.04 | 0.39 | 0.29 | 0.30 | 70.6 | 67.9 |
| S1 | 75 | 0.79 | 0.15 | 1.24 | 4.21 | 1.30 | 2.91 | 0.02 | 0.30 | 56.7 | 66.7 |
| S20 | 75 | 0.79 | 0.15 | 1.08 | 1.51 | 1.12 | 1.24 | 0.20 | 0.30 | 62.7 | 66.7 |
| S17 | 80 | 0.67 | 0.15 | 1.09 | 1.84 | 1.08 | 0.89 | 0.21 | 0.31 | 59.2 | 65.3 |
| S21 | 83 | 0.61 | 0.15 | 0.97 | −0.66 | 0.96 | −0.45 | 0.35 | 0.31 | 64.7 | 64.4 |
| S12 | 88 | 0.49 | 0.15 | 0.93 | −1.55 | 0.95 | −0.61 | 0.39 | 0.32 | 70.1 | 63.6 |
| S2 | 89 | 0.47 | 0.15 | 1.00 | −0.09 | 1.00 | 0.03 | 0.32 | 0.32 | 66.7 | 63.4 |
| S33 | 92 | 0.40 | 0.15 | 1.03 | 0.59 | 1.03 | 0.36 | 0.29 | 0.32 | 62.2 | 63.1 |
| S7 | 96 | 0.31 | 0.15 | 0.99 | −0.12 | 0.98 | −0.31 | 0.33 | 0.32 | 63.2 | 62.8 |
| S13 | 97 | 0.29 | 0.15 | 0.92 | −1.82 | 0.92 | −1.04 | 0.41 | 0.32 | 64.7 | 62.7 |
| S8 | 107 | 0.07 | 0.15 | 0.99 | −0.29 | 1.15 | 2.03 | 0.32 | 0.33 | 63.2 | 63.6 |
| S30 | 114 | −0.09 | 0.15 | 0.96 | −0.80 | 0.95 | −0.67 | 0.37 | 0.33 | 67.2 | 64.9 |
| S3 | 115 | −0.12 | 0.15 | 1.02 | 0.44 | 1.02 | 0.29 | 0.31 | 0.33 | 62.7 | 65.1 |
| S31 | 116 | −0.14 | 0.15 | 0.97 | −0.65 | 0.95 | −0.68 | 0.37 | 0.33 | 66.2 | 65.3 |
| S6 | 118 | −0.18 | 0.15 | 0.88 | −2.47 | 0.85 | −2.18 | 0.47 | 0.33 | 72.1 | 65.7 |
| S10 | 123 | −0.30 | 0.15 | 1.04 | 0.82 | 1.04 | 0.51 | 0.28 | 0.33 | 64.7 | 66.8 |
| S23 | 132 | −0.52 | 0.16 | 1.03 | 0.54 | 1.03 | 0.35 | 0.29 | 0.33 | 68.2 | 69.4 |
| S27 | 136 | −0.62 | 0.16 | 0.99 | −0.07 | 0.94 | −0.67 | 0.35 | 0.33 | 68.7 | 70.7 |
| S4 | 141 | −0.75 | 0.16 | 0.67 | −5.15 | 0.57 | −5.10 | 0.73 | 0.33 | 80.1 | 72.6 |
| S25 | 141 | −0.75 | 0.16 | 1.02 | 0.33 | 1.03 | 0.36 | 0.30 | 0.33 | 73.1 | 72.6 |
| S28 | 146 | −0.89 | 0.17 | 1.02 | 0.30 | 1.05 | 0.48 | 0.30 | 0.33 | 74.1 | 74.7 |
| S24 | 152 | −1.06 | 0.17 | 0.90 | −1.14 | 0.82 | −1.47 | 0.45 | 0.33 | 80.1 | 77.4 |
| S22 | 156 | −1.19 | 0.18 | 0.92 | −0.75 | 0.85 | −1.10 | 0.42 | 0.32 | 80.1 | 79.2 |
| S9 | 173 | −1.84 | 0.22 | 0.95 | −0.32 | 1.19 | 0.96 | 0.32 | 0.31 | 88.1 | 86.9 |
| S5 | 175 | −1.93 | 0.22 | 0.81 | −1.27 | 0.59 | −2.18 | 0.55 | 0.30 | 89.1 | 87.8 |
| S14 | 187 | −2.70 | 0.29 | 0.90 | −0.34 | 1.43 | 1.27 | 0.29 | 0.28 | 94.0 | 93.4 |
| S15 | 196 | −3.89 | 0.47 | 0.82 | −0.33 | 1.14 | 0.43 | 0.31 | 0.21 | 97.5 | 97.5 |
| Mean | 105.5 | 0.00 | 0.18 | 0.99 | −0.05 | 1.06 | 0.25 | | | 72.3 | 72.1 |
| P.SD | 42.9 | 1.25 | 0.15 | 0.11 | 1.56 | 0.29 | 1.88 | | | 9.7 | 8.9 |

The most difficult question based on Table 5 is S16, with a logit measure of 1.85 and a standard error of 0.19. This condition means that it can be 95% certain that the difficulty

level of item 16 lies between two S.E.s below 1.85 and above 1.85, namely 1.47 and 2.23 logit. Then, the easiest question is question S15, with −3.89 logits and a standard error of 0.47 logits. The average value of the item is 0.00 logit, and the standard deviation value of the logit item is 1.25 for the identification of separation (group of items).

The suitability of the SILI knowledge aspect is based on the Outfit MNSQ, Outfit ZSTD, and Pt Mean Corr, which provides information on item misfits or outliers. Based on the appropriate item criteria based on the Outfit MNSQ, there are S35 (2.03) and S18 (2.01), which are misfits because the Outfit MNSQ > 1.5. Meanwhile, based on the Outfit ZSTD, eight questions are misfit, namely S35 (5.93), S18 (4.66), S26 (2.52), S19 (2.91), S8 (2.03), S6 (Outfit −2.18), S5 (−2.18), and S4 (−5.10). Furthermore, based on the Pt Mean Corr, four questions are not misfit, namely questions S34 (−0.2), S13(0.41), S22 (0.42), S24 (0.45), S6 (0.47), and S4 (0.73). Based on the results above, it can be concluded that questions S35 and S18 do not meet the three appropriate question criteria, so S35 and S18 need to be revised. In contrast, the other questions still meet at least one criterion so they can be maintained without revision.

### 3.1.2. Item Characteristics of the SILI Skills Aspect

The item characteristics of the SILI skills aspect are presented in Table 6. The table is sorted from the most difficult to the easiest questions based on the JMLE Measure. The higher the logit value, the more difficult the question item. The most difficult questions are S32, S23, S14, S24, S38, and S13, while the easiest questions are S4, S21, S12, and S3.

**Table 6.** Item characteristics of SILI skills aspects.

| Item | Total Score | JMLE Measure | Model S.E. | Infit | | Outfit | | Pt Measure-Al | | Exact Obs% | Match Exp% |
|------|-------------|--------------|------------|-------|------|--------|------|---------------|------|------------|------------|
| | | | | MNSQ | ZSTD | MNSQ | ZSTD | Corr | Exp | | |
| S32 | 32 | 1.46 | 0.21 | 1.07 | 0.58 | 1.07 | 0.47 | 0.26 | 0.34 | 83.1 | 85.1 |
| S23 | 38 | 1.23 | 0.19 | 0.89 | −0.93 | 0.89 | −0.73 | 0.47 | 0.34 | 85.1 | 82.7 |
| S14 | 44 | 1.02 | 0.18 | 1.03 | 0.29 | 1.08 | 0.66 | 0.30 | 0.35 | 80.6 | 80.4 |
| S13 | 51 | 0.80 | 0.17 | 1.11 | 1.16 | 1.18 | 1.62 | 0.19 | 0.35 | 76.6 | 77.6 |
| S15 | 51 | 0.80 | 0.17 | 1.02 | 0.26 | 1.05 | 0.46 | 0.31 | 0.35 | 78.6 | 77.6 |
| S27 | 51 | 0.80 | 0.17 | 0.93 | −0.72 | 0.92 | −0.69 | 0.43 | 0.35 | 80.6 | 77.6 |
| S5 | 52 | 0.77 | 0.17 | 0.95 | −0.56 | 0.93 | −0.61 | 0.41 | 0.35 | 78.1 | 77.3 |
| S29 | 52 | 0.77 | 0.17 | 1.05 | 0.54 | 1.08 | 0.77 | 0.28 | 0.35 | 77.1 | 77.3 |
| S10 | 53 | 0.74 | 0.17 | 1.05 | 0.55 | 1.11 | 1.04 | 0.27 | 0.34 | 77.6 | 76.9 |
| S31 | 55 | 0.68 | 0.17 | 1.22 | 2.44 | 1.27 | 2.49 | 0.06 | 0.34 | 70.6 | 76.1 |
| S38 | 56 | 0.65 | 0.17 | 0.97 | −0.37 | 1.03 | 0.36 | 0.36 | 0.34 | 80.6 | 75.7 |
| S25 | 57 | 0.62 | 0.17 | 1.06 | 0.73 | 1.09 | 0.98 | 0.26 | 0.34 | 74.1 | 75.3 |
| S35 | 60 | 0.54 | 0.16 | 0.97 | −0.32 | 0.99 | −0.07 | 0.37 | 0.34 | 74.1 | 74.1 |
| S6 | 61 | 0.51 | 0.16 | 0.95 | −0.58 | 0.93 | −0.76 | 0.40 | 0.34 | 74.6 | 73.7 |
| S28 | 62 | 0.49 | 0.16 | 0.92 | −1.11 | 0.92 | −0.94 | 0.44 | 0.34 | 77.1 | 73.3 |
| S2 | 63 | 0.46 | 0.16 | 1.00 | 0.06 | 1.01 | 0.16 | 0.33 | 0.34 | 73.6 | 72.9 |
| S37 | 63 | 0.46 | 0.16 | 1.02 | 0.31 | 1.00 | 0.00 | 0.32 | 0.34 | 69.7 | 72.9 |
| S22 | 64 | 0.44 | 0.16 | 0.93 | −0.96 | 0.90 | −1.17 | 0.43 | 0.34 | 74.1 | 72.5 |
| S26 | 70 | 0.28 | 0.16 | 1.16 | 2.32 | 1.22 | 2.73 | 0.12 | 0.34 | 63.2 | 70.2 |
| S34 | 73 | 0.21 | 0.16 | 1.04 | 0.70 | 1.04 | 0.62 | 0.28 | 0.33 | 68.7 | 69.0 |
| S39 | 75 | 0.16 | 0.15 | 0.94 | −0.95 | 0.94 | −0.93 | 0.40 | 0.33 | 69.2 | 68.3 |
| S19 | 78 | 0.09 | 0.15 | 1.13 | 2.20 | 1.15 | 2.18 | 0.16 | 0.33 | 63.7 | 67.3 |
| S9 | 97 | −0.34 | 0.15 | 1.09 | 2.24 | 1.12 | 1.94 | 0.19 | 0.31 | 57.2 | 62.4 |
| S17 | 99 | −0.38 | 0.15 | 1.10 | 2.49 | 1.17 | 2.65 | 0.16 | 0.31 | 56.2 | 62.0 |
| S20 | 100 | −0.40 | 0.15 | 1.01 | 0.39 | 0.99 | −0.11 | 0.29 | 0.30 | 56.7 | 61.8 |
| S30 | 101 | −0.43 | 0.15 | 0.96 | −1.03 | 1.02 | 0.27 | 0.34 | 0.30 | 69.2 | 61.6 |
| S16 | 102 | −0.45 | 0.15 | 0.94 | −1.51 | 0.93 | −1.07 | 0.38 | 0.30 | 63.7 | 61.5 |
| S18 | 103 | −0.47 | 0.15 | 0.87 | −3.49 | 0.85 | −2.60 | 0.47 | 0.30 | 70.1 | 61.4 |

**Table 6.** *Cont.*

| Item | Total Score | JMLE Measure | Model S.E. | Infit | | Outfit | | Pt Measure-Al | | Exact Obs% | Match Exp% |
|------|------|------|------|------|------|------|------|------|------|------|------|
| | | | | MNSQ | ZSTD | MNSQ | ZSTD | Corr | Exp | | |
| S1 | 107 | −0.56 | 0.15 | 1.00 | 0.07 | 0.96 | −0.65 | 0.30 | 0.30 | 60.2 | 61.1 |
| S33 | 114 | −0.71 | 0.15 | 0.98 | −0.44 | 1.00 | 0.04 | 0.31 | 0.29 | 63.7 | 61.8 |
| S36 | 114 | −0.71 | 0.15 | 0.92 | −2.05 | 0.95 | −0.71 | 0.37 | 0.29 | 71.6 | 61.8 |
| S11 | 128 | −1.03 | 0.15 | 0.88 | −2.69 | 0.84 | −1.85 | 0.42 | 0.26 | 72.6 | 64.6 |
| S8 | 129 | −1.05 | 0.15 | 0.93 | −1.44 | 0.87 | −1.54 | 0.36 | 0.26 | 66.2 | 64.9 |
| S24 | 130 | −1.07 | 0.15 | 0.91 | −1.83 | 0.86 | −1.62 | 0.38 | 0.26 | 64.7 | 65.3 |
| S7 | 134 | −1.17 | 0.16 | 1.00 | −0.05 | 0.99 | −0.09 | 0.26 | 0.25 | 68.7 | 66.9 |
| S3 | 135 | −1.19 | 0.16 | 1.00 | −0.06 | 0.93 | −0.72 | 0.27 | 0.25 | 68.2 | 67.3 |
| S12 | 136 | −1.22 | 0.16 | 0.99 | −0.22 | 0.97 | −0.23 | 0.27 | 0.25 | 68.2 | 67.8 |
| S21 | 137 | −1.24 | 0.16 | 0.91 | −1.64 | 0.84 | −1.58 | 0.38 | 0.25 | 68.7 | 68.2 |
| S4 | 149 | −1.55 | 0.17 | 1.02 | 0.35 | 1.08 | 0.63 | 0.18 | 0.22 | 74.1 | 74.1 |
| Mean | 84.1 | 0.00 | 0.16 | 1.00 | −0.14 | 1.00 | 0.04 | | | 71.1 | 70.4 |
| P.SD | 33.3 | 0.80 | 0.01 | 0.08 | 1.35 | 0.11 | 1.26 | | | 7.1 | 6.6 |

Based on the criteria for the suitability of the question items, the Outfit MNSQ values for all items of the SILI skills aspect are fit items because of the accepted Outfit MNSQ criteria. Meanwhile, based on the Outfit ZSTD, all questions meet the Outfit Z-Standard criteria: $-2.0 < \text{ZSTD} + 2.0$. Furthermore, based on the Pt Mean Corr value, six questions meet the item fit criteria, namely questions S5 (−0.41), S6 (0.40), S39 (0.40), S22(0.43), S27 (0.43), and S28 (0.44). Based on the results above, every question must meet the three appropriate question criteria. Thirty-three questions met the Outfit MNSQ and Outfit ZSTD criteria, while six met three suitability items. Based on the table data, information is obtained that the 39 SILI items for the skills aspect meet the minimum criteria for acceptable items, namely meeting at least one of the Outfit MNSQ or Outfit ZSTD or Pt Mean Corr criteria so that they can be maintained without revision.

### 3.2. Probability of Response SILI

The probability of response to the SILI knowledge aspect is shown in Figure 2a. Meanwhile, the probability of response to the SILI skills aspect is shown in Figure 2b.

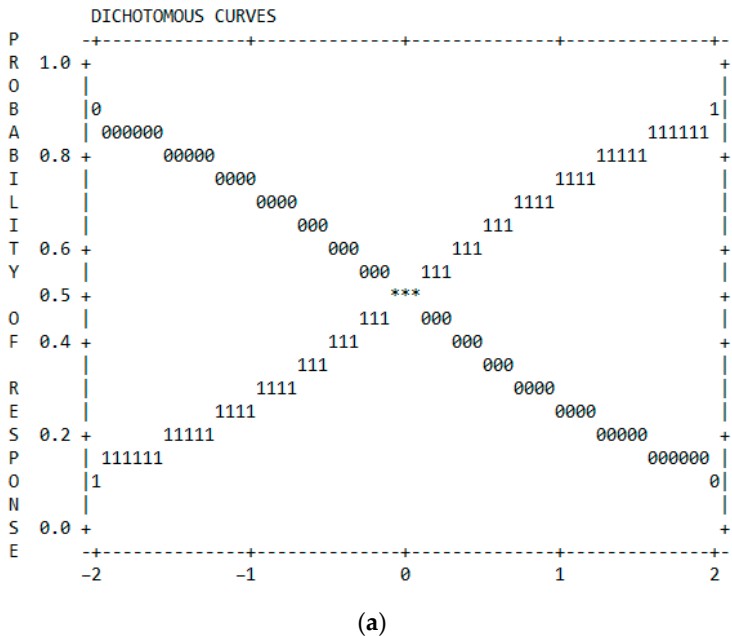

(**a**)

**Figure 2.** *Cont.*

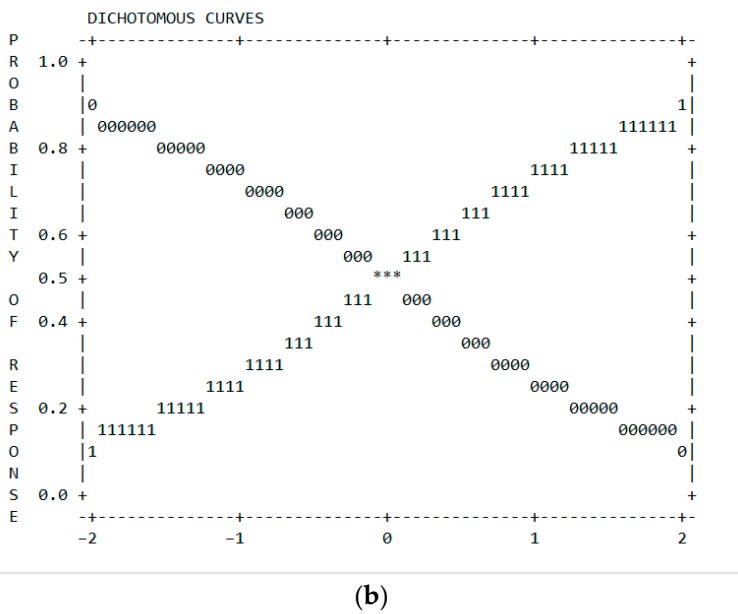

**(b)**

**Figure 2.** (**a**). Probability curve of SILI knowledge aspect. (**b**). Probability curve of SILI skill aspect.

Based on Figure 2a,b, each answer category must have a peak on the curve. Peaks on the curve mean that each category represents a specific portion of each measured aspect, as curves shown.. In the response probability curve for SILI knowledge aspects, two answer categories have the peak of the probability curve as a row of evidence for both answer categories 1 and 0. The *** mark on the curves represents the meeting of two answer categories for prospective science teacher students, namely 1 and 0. Answer category 1 is the probability of the respondent answering correctly, and answer category 0 is the probability of the respondent answering incorrectly. Answer categories 1 and 0 are generally found in every person on the probability curve. This situation means that everyone can answer correctly and wrong. However, the probability of people answering correctly on the SILI knowledge aspects is lower for people with small logit values, while the probability of people answering correctly (answer category 1) is generally found to be higher for people with large logit values, in Figure 2b.

The response probability curve for the SILI skill aspect displays two answer categories with the peak of the probability curve as a row of evidence, both answer categories 1 and 0. In the probability curve, answer categories 1 and 0 are generally found in every person. This condition means that everyone can answer correctly and wrong. However, the probability of people answering correctly on the SILI knowledge aspect is lower for people with small logit values, while the probability of people answering correctly (answer category 1) is generally found to be higher for people with large logit values.

### 3.3. Wright Maps of SILI

The interaction between person and item is displayed as a person-item with a map [71]. Figure 3a shows the person-item SILI data map for the knowledge aspect, and Figure 3b shows the person-item SILI data map for the skills aspect.

Based on Figure 3a, the mean SILI knowledge aspect item response is 0, while the mean person (M) value is 0.2. This means that the respondent's SILI knowledge aspect has an average logit value slightly higher than the average logit item value, so it can be said that respondents generally have a relatively high ability compared to the difficulty of the questions (M position on the second scale line of 0). Based on the logit rule, the best question weights are around M, namely questions S8, S3, S30, S31, and S6, to determine bright students (high ability) from less bright (low ability). There are two students with very high abilities, seven with high abilities, and one with the lowest abilities. Students can answer all questions, so there are no free personal items (none of the respondents

can answer or all respondents can answer; questions that are too difficult and no one can answer; and questions that are too easy, no one can answer). On the correct maps, there are two questions, namely S14 and S15, whose positions are outside the two standard deviations (T) limits, and based on the logit rule, the value is the smallest, so it can be said that this question is very easy.

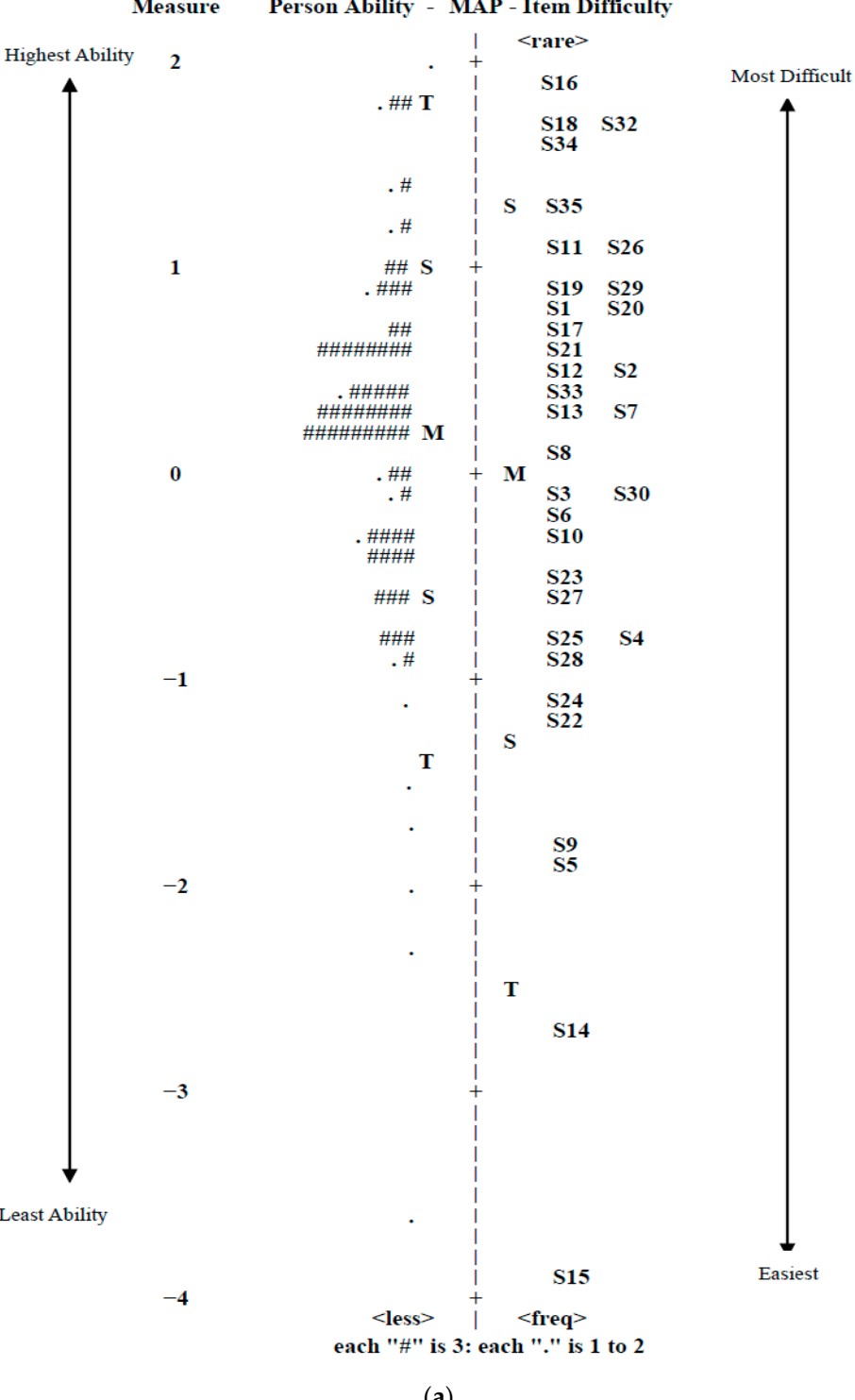

(a)

**Figure 3.** *Cont.*

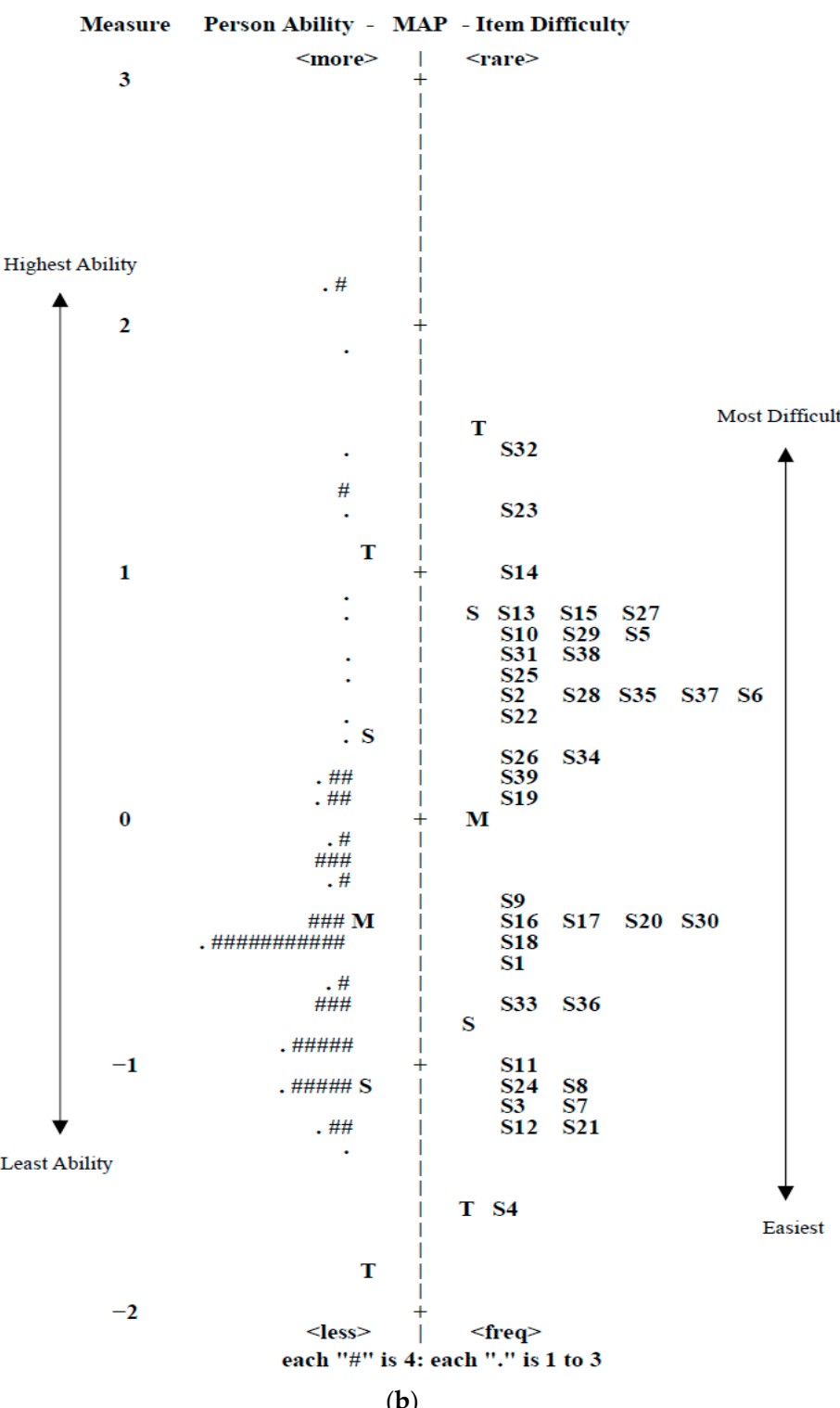

**Figure 3.** (**a**) Wright maps of SILI knowledge aspect. (**b**) Wright maps of SILI skills aspect.

Based on the person-item map, the targeting scale and the difference between prospective science teachers' average ability and LIS items' average difficulty are shown by the difference between the two M values in the person ability and item difficulty sections. The left column M value, namely people's ability, has a logit of 0.19. The M value in the right column is a question difficulty of level 0. So, the targeting scale is 0.19. This value is still tiny on the accepted targeting scale, so there is no targeting error in the SIL knowledge aspect.

Based on Figure 3b, the mean SILI skills aspect item response (M) is 0, while the mean person is −0.4. In the SILI skills aspect, respondents have an average logit value lower than the average logit item value, so respondents generally have an ability lower than average (M position on the four scale lines below 0). The best question weights based on the logit rule are around the mean item (M) with a value of 0.0, namely questions S19, S39, and S26, to determine bright students (high ability). There are seven students with very high abilities, generally some students with medium abilities, and three students with very low abilities.

In the LIS skill aspect, the M value of the left column of the graph map in Figure 3b, namely the average ability of prospective science teachers, has a logit of −0.38. The M value in the right column is the difficulty level of the questions with a logit value of 0. So, the LISI targeting scale for the skills aspect is 0.38. This value is still tiny on the targeting scale received, so there is no targeting error in the SILI skills aspect.

### 3.4. Item Testing: The Dimensionality of the SILI

The results of the SILI dimensionality testing of SILI's aspects are presented in Table 7.

**Table 7.** Dimensionality of SILI.

| | SILI Knowledge Aspect | | | SILI Skills Aspect | | |
|---|---|---|---|---|---|---|
| | Eigen Value | Observed (%) | Expected (%) | Eigen Value | Observed (%) | Expected (%) |
| Total raw variance in observations | 47.9093 | 100 | 100.0 | 49.0036 | 100 | 100 |
| Raw variance explained by measures | 12.9093 | 26.9 | 26.7 | 10.0036 | 20.4 | 20.2 |
| Raw variance explained by persons | 4.4749 | 9.3 | 9.3 | 4.3972 | 9.0 | 8.9 |
| Raw variance explained by items | 8.4343 | 17.6 | 17.5 | 5.6064 | 11.4 | 11.3 |
| Raw unexplained variance (total) | 35.0000 | 73.1 | 100 | 73.3 | 39.0000 | 79.6 | 100 | 79.8 |
| Unexplained variance in 1st contrast | 2.2450 | 4.7 | 6.4 | 2.6907 | 5.5 | 6.9 |
| Unexplained variance in 2nd contrast | 2.0026 | 4.2 | 5.7 | 2.1661 | 4.4 | 5.6 |
| Unexplained variance in 3rd contrast | 1.8961 | 4.0 | 5.4 | 1.8024 | 3.7 | 4.6 |
| Unexplained variance in 4th contrast | 1.7294 | 3.6 | 4.9 | 1.7407 | 3.6 | 4.5 |
| Unexplained variance in 5th contrast | 1.6459 | 3.4 | 4.7 | 1.6395 | 3.3 | 4.2 |

The Table 7 results measuring the row variance of the SILI knowledge aspect are 26.9%. This condition shows that the minimum dimensionality requirement of 20% can be met. Another consideration is the variance that the instrument cannot explain (unexplained variance), ideally not exceeding 15%. Of the five unexplained variances, no unexplained variance exceeds 15%. So, the SILI knowledge aspect item measures what should be measured: students' SILI abilities in the knowledge aspect. Table 7 explains the results of the row variance measurement for the SILI skills aspect, which is 20.4%. This condition shows that the minimum dimensionality requirement of 20% can be met.

Another consideration is the variance that the instrument cannot explain (unexplained variance), ideally not exceeding 15%. Of the five unexplained variances, no unexplained variance exceeds 15%. So, the SILI skills aspect can measure what should be measured. Based on the data above, the SILI knowledge and skills are valid according to the Rasch model measurement.

### 3.5. Detection of Gender Biases

The detection of SILI knowledge aspect and skills aspect items biased towards gender is presented in Figure 4, which shows the result of a probability analysis of men and women.

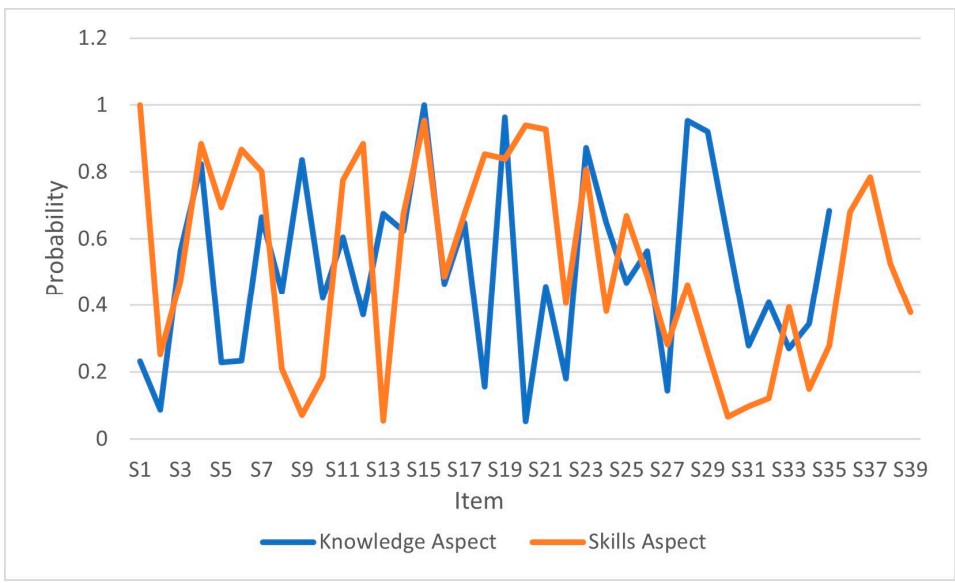

**Figure 4.** Result of probability analysis of SILI Items on gender.

Based on Figure 4, all items of the SILI knowledge aspect and SILI skills aspect meet the probability criteria > 0.05 so that all items can be used without harming either the male or female gender. Meanwhile, for the SILI items on the attitude aspect, all items meet the probability criteria > 0.05 except for items 1, 8, and 30, so a follow-up needs to be conducted so that all items can be used without harming either gender. The gender's ability to answer SILI knowledge aspect items is presented in a graph in Figure 5. Meanwhile, the gender's ability to respond to SILI skills aspect items is shown in Figure 6. Asterisks (green line) indicate the average diff. between the two genders.

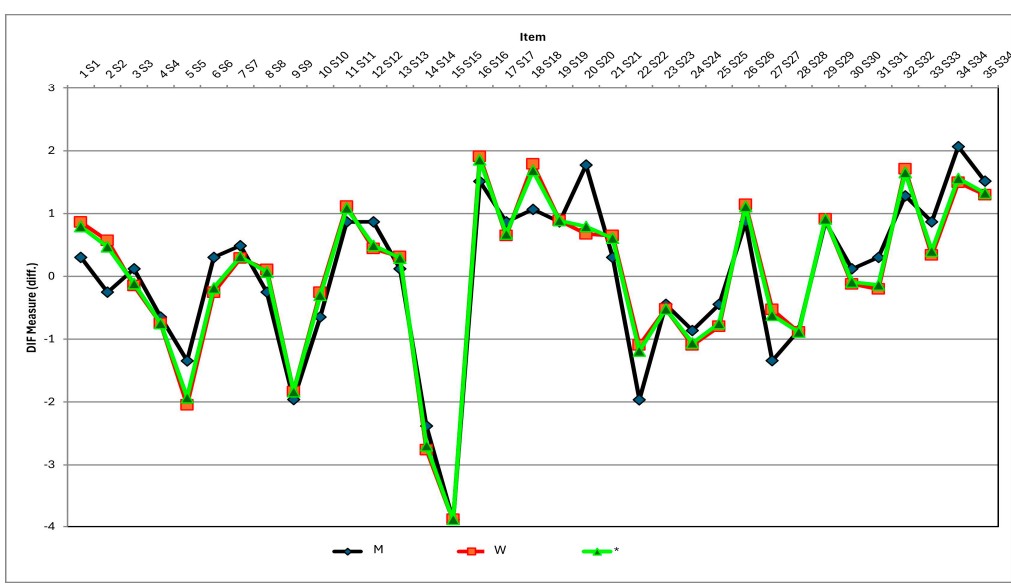

**Figure 5.** Person DIF of gender on SILI knowledge aspects.

Based on Figure 5, the questions that have high difficulty are items 16, 17, and 31, which are easier for men (positions at the bottom 0) to answer than women. Men and women find it equally easy to answer items with a low difficulty (item 15). Meanwhile, for item 5, men find it more difficult to answer than women, while for item 9, women find it easier to answer (below their position) than men; for item 20, men find it more difficult to answer the item, and for item 27, men find it easier to answer than women.

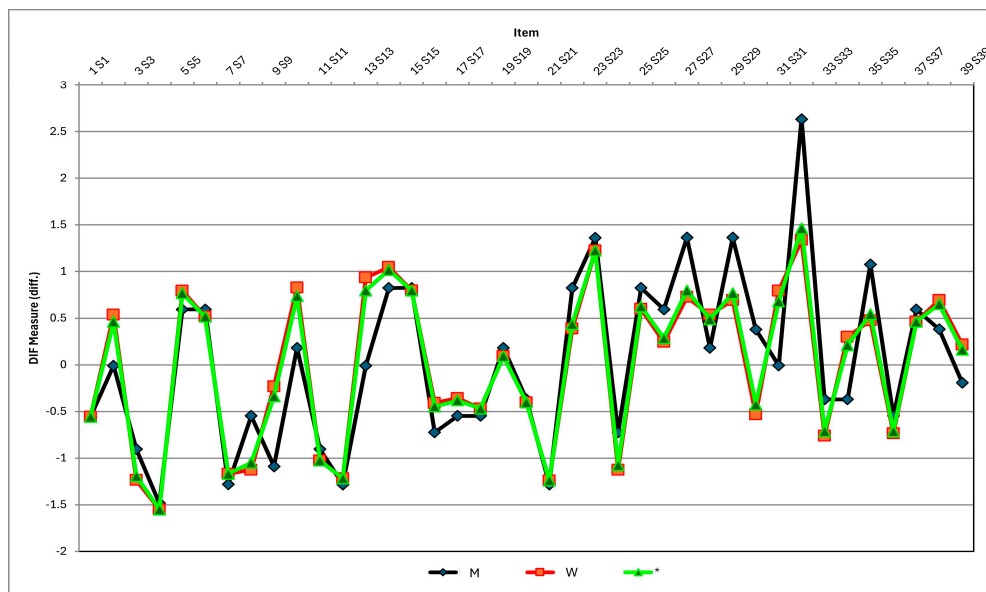

**Figure 6.** Person DIF of gender on skills aspects.

Figure 6 shows that the question that has the highest difficulty is item 32, which is easier for people of the female gender (lower than men). Another difficult item is 23, which both genders can answer with the same ability. Then, items 27, 29, and 35 are easier for the female gender, as seen with the female gender being in the lower position. Meanwhile, items 9 and 12 are easier for men to answer. The easiest items are 4 and 21; both genders are almost the same in their ability to solve the questions.

### 3.6. Detection of Region-Biased

The detection of SILI items that are biased towards regions is presented in Figure 7. The testing criteria are that if the question probability value (Prob) is less than 5%, it indicates that the question item needs to be revised to not harm a particular region.

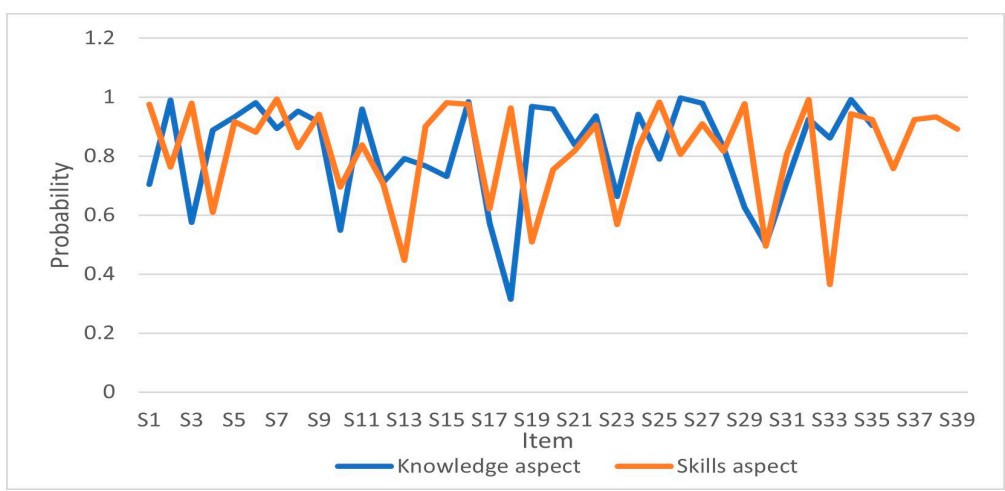

**Figure 7.** Result of probability analysis of SILI items on region.

Based on Figure 7, all items of the SILI knowledge aspect, SILI skills aspect, and SILI attitude aspect meet the probability criteria > 0.05. So, all SILI items, including aspects of knowledge, skills, and attitudes, can be used without harming people from six different regions.

### 3.7. Detection of Science Majors Biases

The detection of biased SILI items towards science majors is presented in Figure 8. The testing criteria are that if the item probability value is less than 5%, it indicates that the item needs to be revised so as to not harm specific science majors.

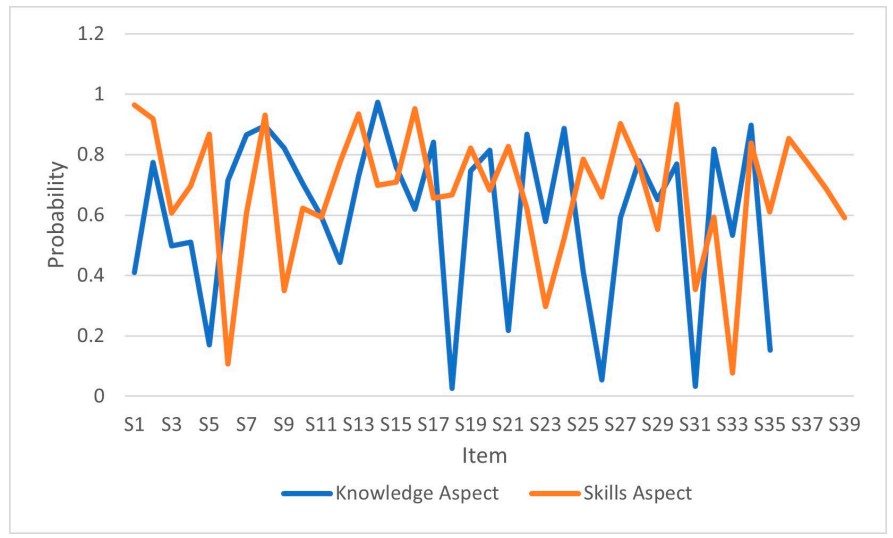

**Figure 8.** Result of probability analysis of SILI items on science majors.

Based on Figure 8, all SILI items meet the probability criteria > 0.05 except items S18 (probability value is 0.0263) and S31 (probability value is 0.0324) on the SILI for the knowledge aspect. So, revisions are needed so all question items can be used without harming people from four different science majors. For more clarity, the ability of the four science groups to answer the SILI on the knowledge aspect is presented in Figure 9.

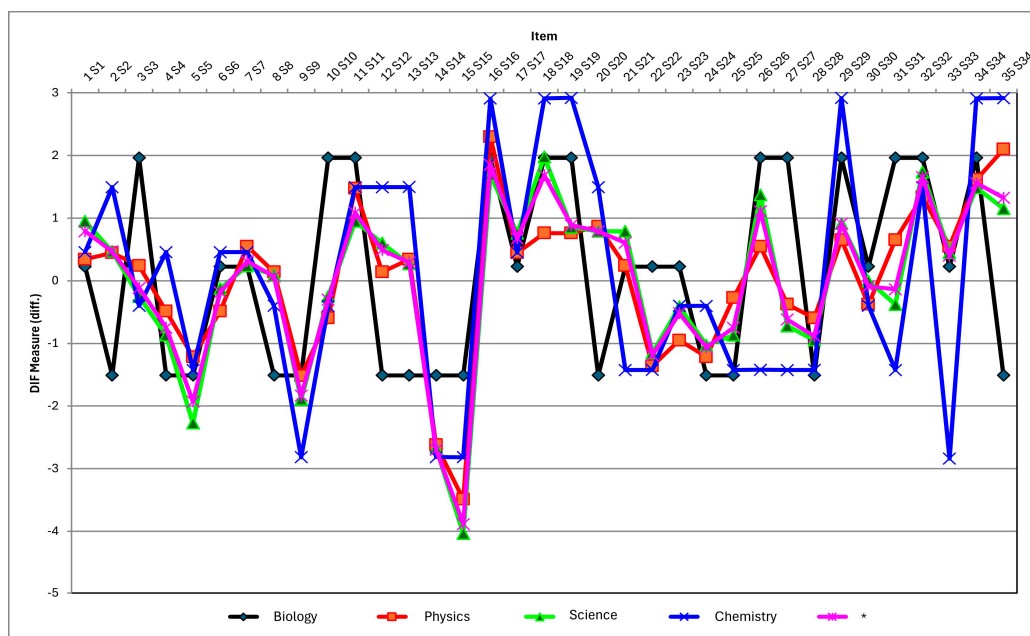

**Figure 9.** Person DIF of science majors on knowledge aspects.

Based on Figure 9, item 18 is difficult for respondents from the chemistry department to answer and easiest for respondents from the physics department. Then, in graph item 31, the question is easy for respondents from the chemistry department to answer and the most difficult for respondents from the biology department to answer.

### *3.8. Information Measurement*

The measurement information graph shows the students' ability level to work on the knowledge aspect of the SILI test. The higher the peak of the information function that can be achieved, the higher the measurement reliability value obtained. The information function for the SILI knowledge aspect is presented in Figures 10 and 11 for the SILI skills aspect.

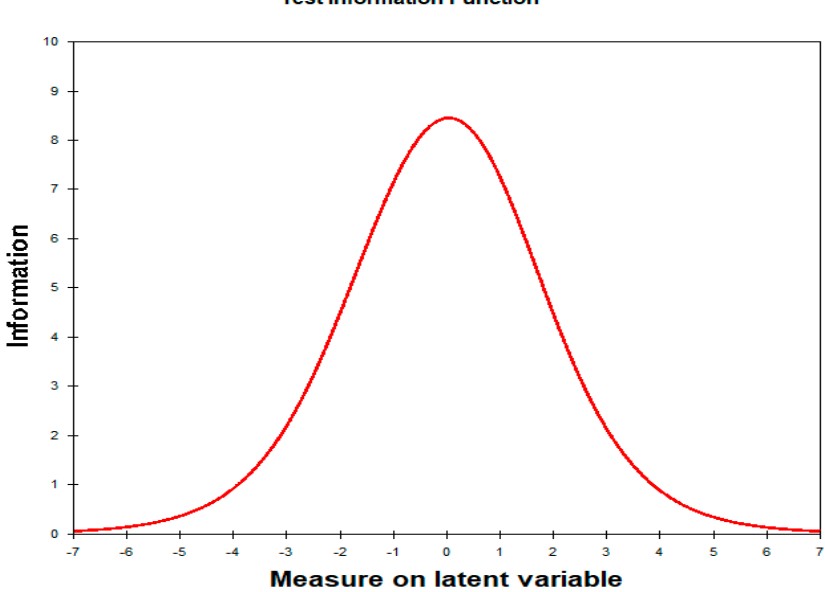

**Figure 10.** Information function for the SILI knowledge aspects.

**Figure 11.** Information function for the SILI skills aspects.

Based on Figure 10, the peak of the curve is at the measurement of the latent variable, which is greater than the value 0. The 35 questions given to 201 students show that the question items are very suitable for knowing the level of the ability of prspective science teachers with slightly higher abilities.

Based on Figure 11, the peak of the curve is at the measurement of the latent variable, which is greater than the value 0. So, the 39 items of the SILI skills aspect given to

201 students show that the items are very suitable for the prospective science teachers who have moderate ability.

All questions are in the fit category based on the Outfit MNSQ value of the SILI knowledge aspect. Then, the results of the SILI aspects of the knowledge analysis item that are biased towards the origin of the science group show that two items do not meet the probability criteria > 0.05 (S18 and S31). So, revisions were made so all question items could be used without harming people from the four science central departments involved. Based on the Outfit MNSQ value for the SILI skills aspect, all questions are in the fit category. Then, the analysis results of items biased towards gender, region, and significance of origin indicate no bias of SILI skills aspect items. The example of the final SILI instrument can be seen in Appendix A for the knowledge aspect and Appendix B for the skills aspect.

## 4. Conclusions

The SILI for evaluating the SIL achievements of prospective science teachers consists of two aspects, namely the knowledge aspect and the skills aspect. The SILI knowledge aspect consists of 18 indicators with 35 multiple-choice questions with four answer choices. SILI skills aspects were developed based on inquiry steps into 37 indicators with 39 multiple-choice questions with five answer choices. Based on the appropriate item criteria based on the Outfit MNSQ, Outfit ZSTD, and PT Mean Corr values, the SILI knowledge aspect of items S35 and S18 does not meet the suitability criteria, so they need to be revised. Meanwhile, in the skills aspect, all SILI items in the skills aspect meet the minimum acceptable item criteria and are maintained without revision. Based on Rasch RMS, the SILI attitude aspect meets all acceptance criteria. Furthermore, there are three items, namely items S8, S4, and S3, which do not meet the Outfit MNSQ, Outfit ZSTD, or PT Mean Corr values' acceptance criteria, so they are referred to as misfit items and are deleted.

The analysis of items biased towards gender shows that all SILI items for the knowledge and skills aspects meet the probability criteria of > 0.05 so that all question items can be used without harming either gender, both male and female. Then, based on the item analysis that is biased towards the region, all SILI items for the knowledge and skills aspects meet the probability criteria > 0.05. So, all SILI items, including knowledge and skills, can be used without harming people from six different regions. Furthermore, based on the analysis items of the SILI that are biased towards science majors, all SILI items meet the probability criteria > 0.05 except for S18 (probability value 0.0263) and S31 (probability value 0.0324) on the SILI for the knowledge aspect. So, revisions were made so all question items could be used without harming people from four different science majors. The dimensionality of the SILI for aspects of knowledge and skills is 26.9% and 20.4%, respectively. Based on all response analyses, it can be concluded that the two aspects of the SILI meet all the accepted item criteria, including the Outfit MNSQ, Outfit ZSTD, and Pt Mean Corr values, item fit, dimensionality, and measurement information without prejudice towards gender, region, and different science majors of prospective teachers. So, 18 knowledge aspect indicators and 37 skills aspect indicators from the SILI are valid according to the Rasch model measurement.

## 5. Limitations and Further Directions

This research has limitations, namely that it was conducted on prospective teachers at several state universities in Indonesia. Then, the sample selection was carried out at several universities, which were taken randomly and did not represent all provinces with different demographics, so there might be bias. Then, the sample categories that took part in this research started from the first year of learning to the final year of the teacher's learning, so this could cause gaps and bias in the data obtained. Therefore, further research can be carried out on prospective teachers in each province to provide more valid and diverse data. Further research was also carried out with a sample of teachers who had experience teaching science in schools, so that scientific inquiry literacy in knowledge and skills could be observed directly.

Another limitation of this research is the multiple-choice questions used in the knowledge and skills aspects. This condition has the potential for prospective teachers to guess, resulting in bias. SILI follow-up research can be modified in short forms to obtain more accurate data. Further research can be carried out to see how students and science teachers achieve SIL for each SIL aspect. This condition will reveal aspects of SIL and SIL indicators that still need to be improved. This condition will give different results for each subject on another demographic profile.

The findings obtained regarding aspects and indicators of the SILI can be used as input for educators in designing SIL instruments for further research. Educators can use SIL indicators according to the needs of the field. Officeholders can also use it to evaluate science learning by the nature of science.

**Author Contributions:** Conceptualization, D.R.D., A.S. (Andi Suhandi), I.K., A.S. (Achmad Samsudin) and F.C.W.; methodology, D.R.D. and A.S. (Andi Suhandi); software, D.R.D.; validation, D.R.D., A.S. (Andi Suhandi), I.K., A.S. (Achmad Samsudin) and F.C.W.; formal analysis, D.R.D., A.S. (Andi Suhandi) and I.K.; investigation, D.R.D. and A.S. (Andi Suhandi); resources, F.C.W.; data curation, A.S. (Achmad Samsudin); visualization, D.R.D. and F.C.W.; writing—original draft, D.R.D. and A.S. (Andi Suhandi); writing—review and editing, D.R.D., A.S. (Andi Suhandi), I.K., A.S. (Achmad Samsudin) and F.C.W.; funding acquisition, D.R.D. All authors have read and agreed to the published version of the manuscript.

**Funding:** This research was funded by BPPT and LPDP grant number 0869/J5.2.3/BPI.06/10/2021 And the APC was funded by BPPT and LPDP.

**Institutional Review Board Statement:** This study was conducted in accordance with the Declaration of Helsinki and approved by the Ethics Committee of Universitas Negeri Jakarta (protocol code 202/UN39.14/PT.01.05/III/2024 dated 8 March 2024).

**Informed Consent Statement:** Informed consent was obtained from all subjects involved in the study.

**Data Availability Statement:** Data was contained within the article. All individuals included in this section have consented to the acknowledgement.

**Acknowledgments:** Thank you to those who have helped carry out this research. Thanks to BPPT and LPDP as the parties who funded this research and publication. Thank you to lecturers and students of prospective science teachers at UNJ, UNTIRTA, UIN SUSKA, UNP, UNSIL, and IAIN Kudus for participating in this research. Thank you to Iwan Sugihartono, M.Si, LPPM UNJ, and LPPM UPI, who helped with this research. Thank you, UPI, for supporting and facilitating this research.

**Conflicts of Interest:** The author declares no conflict of interest.

### Appendix A. Example of the SILI Knowledge Aspect (Indicator Number: 1, 10, 11, 12, 13, 17, 21, 22, and 23)

1. To learn about nature, scientists must study the world in an organized way. They use skills known as. . .

   a. Scientific Method    b. Observation    c. Inquiry
   d. Experiment    e. Hypothesis

10. The following is not considered a tool for a physics experimental activity. . .

    a. Nichrome Wire    b. Cable    c. Crocodile Claws
    d. Power supply    e. Ammeter

11. The following are not classified as materials for a physics experimental activity. . .

    a. Nichrome Wire    b. Cooking oil    c. Rubber
    d. Salt    e. Bunsen burner

12. A physical quantity whose value also changes when the value of another quantity is modified is called a(n). . .

    a. Independent variable    b. Control variable    c. Accompanying variable
    d. Dependent variable    e. Impacted variable

13. A physical quantity whose value is varied to determine variations in other quantities in an experiment is called a(n). . .

a. Independent variable  b. Control variable  c. Accompanying variable
d. Dependent variable  e. Impacted variable

17. Look at the three pictures below. Figure I informs Indonesia's Per Capita Income 2006–2011, Figure II tells the mass of grade 4 elementary school students, and Figure III informs the education level of the parents of science students.

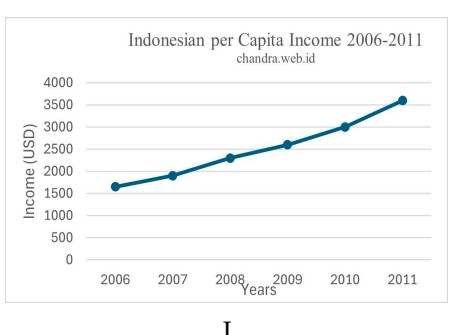

I

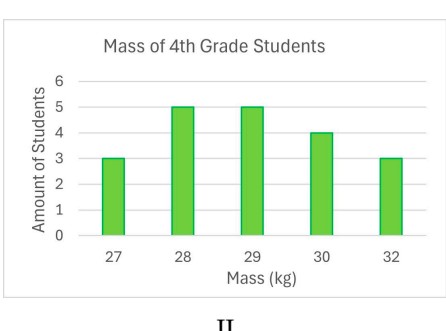

II

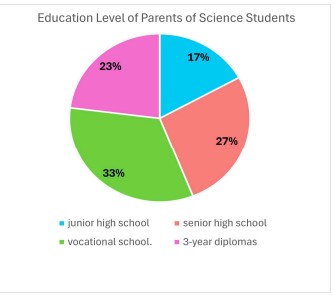

III

From the third image above, which one is a graph? . . .

a. I only  b. II only  c. III only  d. I and II  e. I and III

21. Scientists develop scientific procedures to make discoveries, test hypotheses, or demonstrate known facts. This procedure can be a lot of fun, but scientists need to make sure to control variables when they perform a(n). . .

a. Inquiry  b. Scientific method  c. Observation
d. Experiment  e. Measurement

22. What skills do you use when you sort items into groups?

a. Classifying  b. Observing  c. Measuring
d. Predicting  e. Generalizing

23. Which skill uses your five senses?

a. Predicting  b. Measuring  c. Communicating
d. Observing  e. Experimenting

**Appendix B. Example of the SILI Skills Aspect (1, 2, 6, and 8)**

1. A scientist wants to investigate how a scientific phenomenon can occur through a series of scientific experimental processes consisting of the following non-sequential steps:

    1. Draw conclusions based on data analysis.
    2. Put forward a prediction.
    3. Provide assumptions.
    4. Produce explanations of phenomena.

Order the steps from first to last that scientists take to obtain scientific answers about this phenomenon:

(a)  2, 3, 1, 4
(b)  1, 3, 4, 2
(c)  4, 2, 3, 1
(d)  3, 4, 2, 1

2. A science student heats a beaker containing water mixed with ice and observes the temperature using a thermometer. While continuing to stir the mixture of water and ice and after heating it for a while, it turned out that the temperature of the water mixed with ice did not rise. He concluded that the heat provided did not raise the temperature of the water for a long time because it was continuously stirred. What kind of opinion is needed to correct this conclusion?

    (a)  Nothing; it is an acceptable conclusion, as stated.
    (b)  It would be a correct conclusion only if there was no ice during the process.

(c)     It would be correct if the experiment started with ice only.
(d)     It would be correct if there was no stirring.

6.     A physics student uses a spring balance to measure the weight of several weights, each of which has a mass specified by the manufacturer. After obtaining the data, the student then makes a graph and finds the most suitable trend line based on the data obtained.

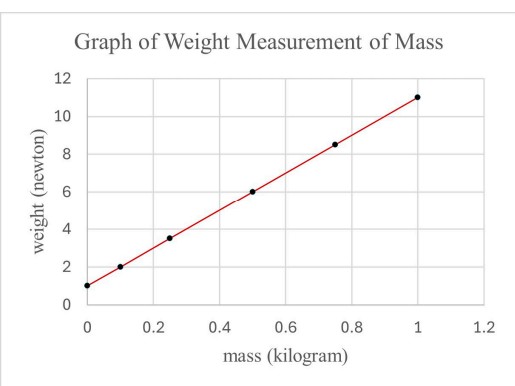

Based on the trend of the curve, the student then stated that "when the mass of an object is 0, then the object will weigh 1 newton".

Do you agree or not with the student's statement? And state the reasons why you agree or disagree.

(a)     Yes, I agree because the graphic data shows that.
(b)     Yes, agree, students have discovered something new.
(c)     Disagree, students make predictions outside the range of the data incorrectly.
(d)     Disagree, this situation is physically impossible.

8.     In investigating question 6 above, what is the most likely reason why the trend line obtained does not pass through the origin (0, 0) on the graph?

(a)     The spring balance scale used is less accurate.
(b)     The balance scale is not adjusted to read 0 when there is no weight attached to it.
(c)     The spring balance has a spring that is no longer elastic.
(d)     The mass label on the load is incorrect.

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
