# Peer review of "Development and Validation of Scientific Inquiry Literacy Instrument (SILI) Using Rasch Measurement Model"

_education, doi:10.3390/educsci14030322_

Round 1

Reviewer 1 Report

Comments and Suggestions for Authors

The article presents an exhaustive statistical analysis that explains the validity of the items used in the questionnaires. The problem is that it does not explain the questionnaires nor can one observe the type of questions to which the students respond. As it says at the end of the article, students could respond to many of the answers given in the way they consider socially more appropriate for their professional future as teachers. The article explains the whole process of designing the instrument, but the qualitative part is taken for granted without actually showing the final instrument of questions. In my opinion, it would be important to disclose the questions used in the questionnaires or the reason why they are not considered relevant for teaching. Statistical data are very important, but without seeing the questions it is possible that students answer in a similar way and we do not have an interesting instrument for the investigation of knowledge, skills and attitudes. If this part can be improved it seems to me that the rest of the article is very good (there are even statistical results that can be removed to improve this part if they need space).

Author Response

Response to Reviewer X Comments

1. Summary

2. Questions for General Evaluation

Reviewer’s Evaluation

Response and Revisions

Does the introduction provide sufficient background and include all relevant references?

Yes/Can be improved/Must be improved/Not applicable

I have revised it by adding some information to the introduction and rearranging the position of paragraphs that are not suitable. My corresponding response is presented in the point-by-point response letter, the same as below.

Are all the cited references relevant to the research?

Yes/Can be improved/Must be improved/Not applicable

I have revised it by adding 15 references relevant to the research. Additional references can be seen in the bibliography.

Is the research design appropriate?

Yes/Can be improved/Must be improved/Not applicable

Revisions are made by explaining the appropriate design in more detail by looking at Figure 1 to explain the appropriate research design.

Are the methods adequately described?

Yes/Can be improved/Must be improved/Not applicable

Revisions have been made to the Sequential Exploratory Research Procedure on the Development of SILI, presented in Figure 1, and the method explanation has been added on Page 4, lines 176-190.

Addition of sections 2.3 and 2.4 to become part of section 2. Materials and methods. Then add information on page 6, lines 230-234 and 242-234.

Added data collection information on page 6, lines 267-271, 283-284,

I added an explanation of analysis and interpretation on pages 9 to 10, lines 292-306. Then Added an explanation section about person and item reliability, dimensionality, Functionality of the Response Categories, Differential Item Functioning (DIF), Targeting, and Rasch Rating Scale Method (RSM) on pages 10-11.

Are the results clearly presented?

Yes/Can be improved/Must be improved/Not applicable

A revision has been carried out to explain the indicators and aspects of the resulting SILI by adding information to clarify the findings.

Revisions are carried out by correcting typos, writing errors, and sentences that could be better structured. It can be seen on page 12, lines 409-410, Table 6, page 16, lines 499-500, and page 17.

I added an explanation of Rasch RMS on page 19, lines 564-577.

I added section 3.3. Wright Maps SILI is on page 19, line 578.

I Improved the display of the wrigh map by adding captions and arrows for easy reading on pages 20-21.

I added the Wright maps explanation on lines 620-638.

I revised section '3.4. Detection of Gender-Biased', Table 10 is simplified into Figure 5 and adds some information in lines 673-674.

I revised section '3.4. Detection of Gender-Biased', Table 10 is simplified into Figure 5 and adds some information in lines 673-674.

I revised section '3.5. For detecting "region-biased"', Table 11 is simplified into Figure 9.

In section 3.5, Table 12 is simplified into Figure 10.

Revisions have been made by adding examples of questions used in appendices 1, 2, and 3.

Revisions were made by adding Rasch RMS information on page 30, lines 844-863.

Revisions were made by adding information about probability curves on page 31, lines 871-873 and 833-886.

Are the conclusions supported by the results?

Yes/Can be improved/Must be improved/Not applicable

Revisions were made to cunculation and limitation by removing repeated sentences in line 895 and improving the conclusion by adding lines 900-903.

3. Point-by-point response to Comments and Suggestions for Authors

Comments 1:
The article presents an exhaustive statistical analysis that explains the validity of the items used in the questionnaires. The problem is that it does not explain the questionnaires nor can one observe the type of questions to which the students respond. As it says at the end of the article, students could respond to many of the answers given in the way they consider socially more appropriate for their professional future as teachers.

Response 1: Thank you for pointing this out. I agree with this comment. I have added examples of the SILI instrument for knowledge aspects in Appendix 1, skills aspects in Appendix 2, and attitude aspects in Appendix 3, pages 35-37. Apart from that, I also made several other revisions, namely:

- I also added section ‘2.3. Research samples on page 5 to clarify samples and participants in the qualitative and quantitative research sections. Initially, on page 8, section 4.2, participants were revised to section "2. Materials and Methods". Participant information is combined with the sample section on page 5, section ‘2.3. Research samples’.

- Section 2.3 was initially 2.3. Measurement Test (only explains quantitative research) Changed to 2.4. Research instruments that explain the instruments used in qualitative and quantitative research are presented on page 6, with additional descriptions for prospective science students.

- I also added additional explanations in section 2.5. Data collection on qualitative and quantitative research.

- Revised Data collection on page 9, lines 267-278

Comments 2:

The article explains the whole process of designing the instrument, but the qualitative part is taken for granted without actually showing the final instrument of questions. In my opinion, it would be important to disclose the questions used in the questionnaires or the reason why they are not considered relevant for teaching.

Artikel tersebut menjelaskan keseluruhan proses perancangan instrumen, namun bagian kualitatif dibiarkan begitu saja tanpa benar-benar menunjukkan instrumen pertanyaan akhir. Menurut pendapat saya, penting untuk mengungkapkan pertanyaan-pertanyaan yang digunakan dalam kuesioner atau alasan mengapa pertanyaan-pertanyaan tersebut dianggap tidak relevan untuk pengajaran.

Response 2: Thank you for pointing this out. I Agree. Some of the revisions I made were:

-I revised ‘section 2.6. Analysis and Interpretation by clarifying qualitative data analysis on pages 9-10, 291-307. Initially, data analysis was not presented in Section '2. Materials and Methods only show the quantitative section in the introduction and are incomplete.

- The revision was made by adding section 2.6. Analysis and Interpretation to explain data analysis carried out in qualitative and quantitative research on pages 9-10. 291-307.

Then, I revised the appropriate design in more detail by looking at Figure 1 to explain the appropriate research design.

-Revisions have been made to the Sequential Exploratory Research Procedure on the Development of SILI, presented in Figure 1, and the method explanation has been added on Page 4, lines 176-190.

-Sections 2.3 and 2.4 were added to become part of section 2. Materials and methods. Then add information on page 6, lines 230-234 and 242-234.

-Added data collection information on page 6, lines 267-271, 283-284,

Comments 3:

Statistical data are very important, but without seeing the questions it is possible that students answer in a similar way and we do not have an interesting instrument for the investigation of knowledge, skills and attitudes. If this part can be improved it seems to me that the rest of the article is very good (there are even statistical results that can be removed to improve this part if they need space).

Response 3: Thank you for pointing this out. I Agree. Some of the revisions I made are:

- I have revised this by deleting 'Table 10, Results of Differential Item Functioning (DIF) analysis of SILI items' and presenting the data in Figure 5, page 25, lines 66-68. Then I deleted 'Table 11, Results of Differential Item Functioning (DIF) analysis of SILI items' and presented the data in Figure 9. pages 26-27, lines 729-737. Then I deleted 'Table 12, Results of Differential Item Functioning (DIF) analysis of SILI items' and presented the data in Figure 10, page 27, lines 742-748.

- I have added examples of the SILI instrument for knowledge aspects in Appendix 1, skills aspects in Appendix 2, and attitude aspects in Appendix 3, pages 35-37.

4. Response to Comments on the Quality of English Language

Point 1:

-

5. Additional clarifications

-

Reviewer 2 Report

Comments and Suggestions for Authors

Dear authors,

Section 1. Introduction is a summary of the beneficial features at science education programs, especially in their ability to improve thinking and investigating methods. Scientific Inquiry Literacy (SIL) in its tool version such as SILI (Scientific Inquiry Literacy Instrument) is then introduced together with VOSI (Views of Scientific Inquiry) to highlight their capability to measure scientific inquiry. The first instrument is applied to this research to target knowledge, skills, and attitudes using the Rasch model with Winsteps 5.4.1. application.

Several short sentences are then produced probably with chat GPT enlist the goodness of such instruments for measuring scientific inquiries. See lines 63-78 and lines 79-103.

At line 104 forwards the authors introduce the Rasch model which is an instrument that develops skills for research beyond the Classical Test Theory (CTT). The Rasch model has in fact the chance to convert ‘ordinal responses’ into ratios which are more employable when studying probability. An item map can be created which includes the classification of the calculation of questions items and people. Therefore, test takers’ probabilities are correctly predicted. Another advantage is the Rasch model predicts missing data on systematic response patterns.

The focus addresses then the production of prospective science teachers in Indonesia. A list of four different study programs in physics, chemistry, biology, and science, is then offered by authors at line 198-204. SILI is indicated as a necessary instrument to evaluate students’ abilities in scientific inquiry.

The SILI is composed of 35 items on knowledge features, 39 skills features, and 30 questionnaires on attitude features on research. Table 1 at line 217 evaluates concepts that support SIL knowledge. Table 2 at line 227 evaluates skills features at SIL. Attitudes are then exemplified in Table 3, line 232. In the chosen sample, 201 prospective science teachers in the above-mentioned four study programs take part in the research by authors. Their descriptive demographic profile is developed in Table 4 at line 244, showing a. gender, b. year at university, c. region of origin, and d. major at university. Time frame for research was October 2022 until June 2023, and the questionnaire was validated using the Rasch model with Winsteps 5.4.1. application. See sub-section 2.5. Data Collection, lines 251-70. Specific attention is provided to the JMLE (Joint Maximum Likelihood Estimation) calculated through the three features of a. Outfit mean square (MNSQ), b. Outfit Z-Standard value (ZSTD), and c. Point measure correlation value (Pt Mean Corr). Cronbach’s alpha (KR20) is also calculated to quantify the level of agreement on standardized scales. Five experts validated the instrument of SILI from the qualitative point of view. See lines 180-89. Table 5 at line 278 recapitulates the KR20, and the other JMLE measurements. Many descriptive outcomes on the JMLE measurements are detailed at lines 279-313.

Sub-section 3.1 Item Characteristics, presents Tables 6–7–8, which are measurement of the three main SILI features (knowledge, skills, and attitudes) from which the authors have drawn answers on the questionnaires. Infit and Outfit measurements are then showed for the respective three SILI features. Figure 2 at line 381 shows curve of items that present deviation (in blue) from the ICC (Item Characteristic Curve) and IRF (Item Response Function) models are ideal lines (in red). Figure 3.a 3.b, and 3.c at lines 401-407 indicate respectively the probability curve for the three main SILI features inquired by authors. A detailed discussion on the probabilities is then shown on lines 408-33. The following three figures Figure 4.a, 4.b, and 4.c show Wright maps on the results of the dichotomous item response model used by authors with the target at examining the strengths and weaknesses in their measurements in terms of item distribution. A detailed discussion is then provided at lines 440-69. Dimensionality of SILI is also shown in Table 9 at line 489, indicating a high percentage of ‘raw unexplained variance in total’ in the data. Eigen value for the ‘total raw variance in observations’ is at a middle point of the expected percentage in all the three main features, they are now abbreviated as ‘Knowledge Aspects (KA)’, ‘Skills Aspects (SA)’, and Attitude Aspects (AA)’. These abbreviations are useful to better organize outcomes in the Differential Item Functioning (DIF) analysis are shown in Table 10 at line 494. Gender bias is detected on three items for the AA, so that the following figures Figure 5–6–7 show graphically whether the biases are larger than the authors’ accepted thresholds for analysis. Table 11 shows the Chi-square at DIF of SILI items on region with probabilities between-Class/Group. Same is done in Table 12 on science majors. Figure 9– 10–11, show the respective curves of latent variables in the three main features at research. In their follow-up analysis the authors elucidate on adjustments have been made with the aim at reducing biases in the SILI analysis. Tables 13–14–15 are also to show how the authors have reduced misfit items. Conclusions and limitations for future research indicate the multiple choice instrument for questionnaire can condition respondents to guess on answers not conducting to an effective evaluation of the inquired features under study.  

References for the review:

[1] Boone, W.J., Staver, J.R. (2020). Wright Maps (Part 3 and Counting...). In: Advances in Rasch Analyses in the Human Sciences. Springer, Cham. https://doi.org/10.1007/978-3-030-43420-5_16  

[2] Cantó-Cerdán, M., Cacho-Martínez, P., Lara-Lacárcel, F. et al. (2021). Rasch analysis for development and reduction of Symptom Questionnaire for Visual Dysfunctions (SQVD). Sci Rep 11, 14855. https://doi.org/10.1038/s41598-021-94166-9

[3] DiStefano, C., Jiang, N. (2020). Applying the Rasch Rating Scale Method to Questionnaire Data. In: Khine, M. (eds) Rasch Measurement. Springer, Singapore. https://doi.org/10.1007/978-981-15-1800-3_3

[4] Lutsenko, M., Seytmanbitov, D. (2022). Calculation of Knowledge Levels in the Rush Model (Methods, Examples). In: Lyapin, A., Kalinina, O. (eds) Digital Technologies in Teaching and Learning Strategies. DTTLS 2021. Lecture Notes in Information Systems and Organisation, vol 56. Springer, Cham. https://doi.org/10.1007/978-3-031-05175-3_3

Advised amendments:

I advise you to provide a more detailed and reasoned paper, emphasizing the need for clear and fluid research targets beyond just the use of Winsteps. Here are a couple of suggestions drawn from the above referenced papers:

¨      Integration of Wright Maps:

Consider commenting on Wright Maps, as discussed in Boone and Staver (2020) [1], to enhance the presentation of item characteristics and measurement in your study.

¨      Application of Rasch Analysis Techniques:

Explore additional Rasch analysis techniques discussed in relevant papers. For instance, Cantó-Cerdán et al. (2021) [2] applied Rasch analysis for the development and reduction of a Symptom Questionnaire, showcasing the versatility of Rasch models in various contexts. Considering the specific context of your study, explore how similar techniques or adaptations could be applied to further refine and validate your instrument.

¨      Consideration of Rating Scale Method:

DiStefano and Jiang (2020) [3] discuss the application of the Rasch Rating Scale Method to questionnaire data. Consider incorporating insights from this paper to strengthen the analysis of the attitude features in your study. This could provide additional depth in understanding respondents’ attitudes towards scientific inquiry, leveraging the Rasch Rating Scale Method for more nuanced insights.

¨      Knowledge Level Calculation in the Rasch Model:

Lutsenko and Seytmanbitov (2022) [4] discuss the calculation of knowledge levels in the Rasch model. Explore the methods and examples presented in this paper to enhance the analysis of knowledge features in your study. This may provide additional insights into how knowledge levels are distributed across different aspects of the scientific inquiry, contributing to a more comprehensive understanding.

These suggestions aim to enrich your study by incorporating insights and methodologies from related papers, ensuring a robust and well-rounded application of Rasch analysis techniques to your specific context.

Other amendments:

1.      What is the ‘NoS’ mentioned at line 32 ? Please clear to a non-expert audience,

2.      There is one value with comma in Table 5, line 278, last but not least on the right in first row. Please uniform to the other values they are with points,

3.      Please check at line 293 whether in Anglo-Saxon style after semicolon you should start with a small letter instead that with a capital letter (…of 1.69; The value of the SILI person…),

4.      Please check and correct in third column first row at Table 6 line 321 ‘JMLE masure’,

5.      If I were you, I would put paragraph lines 383-390 before Figure 2.a, 2.b, and 2.c, so that I could learn before what the acronyms ICC (Item Characteristic Curve) and IRF (Item Response Function) respectively mean. Think also to renumber figures with letters ‘a’ to ‘c’ as you did in the case of Figure 3.a 3.b, and 3.c,

6.      Figure 3.a 3.b, and 3.c are indicated differently at line 408 of the main body (they are without point between number and letter). Please choose one only method to indicate figures along the text,

7.      The acronym of Differential Item Functioning (DIF) is already solved in Table 10 at line 494, so that you do not need to solve it again in Table 11 at line 575, as you already did in the previous Figure 5–6–7,

8.      What is the asterisk at line 600 in Figure 8? Does this indicate that a residual category is to be calculated?

9.      Please mention reference no. 5 in the following way:

Yannier, N., Hudson, S. E., Koedinger, K. R., Hirsh-Pasek, K., Golinkoff, R.M., Munakata, Y. et al. (2021). Active learning: "Hands-on" meets "minds-on". Science. Oct;374(6563):26-30. doi: 10.1126/science.abj9957. Epub 2021 Sep 30. PMID: 34591619.

10.   Please mention reference no. 23 in the following way:

Brownscombe, J. W., Lédée, E. J. I., Raby, G. D. et al. (2019). Conducting and interpreting fish telemetry studies: considerations for researchers and resource managers. Rev Fish Biol Fisheries 29, 369–400. https://doi.org/10.1007/s11160-019-09560-4,

11.   Reference no. 38: the dot after the year of publication between parentheses is missing,

12.   Please mention reference no. 51 in the following way:

Dahlgren A, Semakula D, Chesire F et al. Critical thinking about treatment effects in Eastern Africa: development and Rasch analysis of an assessment tool [version 1; peer review: awaiting peer review]. F1000Research 2023, 12:887. https://doi.org/10.12688/f1000research.132052.1.

With Best Regards,

Author Response

Response to Reviewer X Comments

1. Summary

2. Questions for General Evaluation

Reviewer’s Evaluation

Response and Revisions

Does the introduction provide sufficient background and include all relevant references?

Yes/Can be improved/Must be improved/Not applicable

I have revised it by adding some information to the introduction and rearranging the position of paragraphs that are not suitable. My corresponding response is presented in the point-by-point response letter, the same as below.

Are all the cited references relevant to the research?

Yes/Can be improved/Must be improved/Not applicable

I have revised it by adding 15 references relevant to the research. Additional references can be seen in the bibliography.

Is the research design appropriate?

Yes/Can be improved/Must be improved/Not applicable

Revisions are made by explaining the appropriate design in more detail by looking at Figure 1 to explain the appropriate research design.

Are the methods adequately described?

Yes/Can be improved/Must be improved/Not applicable

Revisions have been made to the Sequential Exploratory Research Procedure on the Development of SILI, presented in Figure 1, and the method explanation has been added on Page 4, lines 176-190.

Addition of sections 2.3 and 2.4 to become part of section 2. Materials and methods. Then add information on page 6, lines 230-234 and 242-234.

Added data collection information on page 6, lines 267-271, 283-284,

I added an explanation of analysis and interpretation on pages 9 to 10, lines 292-306. Then Added an explanation section about person and item reliability, dimensionality, Functionality of the Response Categories, Differential Item Functioning (DIF), Targeting, and Rasch Rating Scale Method (RSM) on pages 10-11.

Are the results clearly presented?

Yes/Can be improved/Must be improved/Not applicable

A revision has been carried out to explain the indicators and aspects of the resulting SILI by adding information to clarify the findings.

Revisions are carried out by correcting typos, writing errors, and sentences that could be better structured. It can be seen on page 12, lines 409-410, Table 6, page 16, lines 499-500, and page 17.

I added an explanation of Rasch RMS on page 19, lines 564-577.

I added section 3.3. Wright Maps SILI is on page 19, line 578.

I Improved the display of the wrigh map by adding captions and arrows for easy reading on pages 20-21.

I added the Wright maps explanation on lines 620-638.

I revised section '3.4. Detection of Gender-Biased', Table 10 is simplified into Figure 5 and adds some information in lines 673-674.

I revised section '3.4. Detection of Gender-Biased', Table 10 is simplified into Figure 5 and adds some information in lines 673-674.

I revised section '3.5. For detecting "region-biased"', Table 11 is simplified into Figure 9.

In section 3.5, Table 12 is simplified into Figure 10.

Revisions have been made by adding examples of questions used in appendices 1, 2, and 3.

Revisions were made by adding Rasch RMS information on page 30, lines 844-863.

Revisions were made by adding information about probability curves on page 31, lines 871-873 and 833-886.

Are the conclusions supported by the results?

Yes/Can be improved/Must be improved/Not applicable

Revisions were made to cunculation and limitation by removing repeated sentences in line 895 and improving the conclusion by adding lines 900-903.

3. Point-by-point response to Comments and Suggestions for Authors

Comments 1:
Integration of Wright Maps:

Consider commenting on Wright Maps, as discussed in Boone and Staver (2020) [1], to enhance the presentation of item characteristics and measurement in your study.

Response 1: Thank you for pointing this out. I agree with this comment. Therefore, I have made revisions by giving a header on the person side; use a title in caps PERSON ABILITY and a title on the item side. ITEM DIFFICULTY. Then, two arrow lines are added on both sides to indicate the level of ability and item difficulty. At the end of the two arrows, use the phrases “highest ability” and “least ability” on the person side and “Most difficult” and “easiest” on the item difficulty side. This revision was made to all three Wright Maps. It can be seen on pages 20 and 21.

Comments 2:

Application of Rasch Analysis Techniques:

Explore additional Rasch analysis techniques discussed in relevant papers. For instance, Cantó-Cerdán et al. (2021) [2] applied Rasch analysis for the development and reduction of a Symptom Questionnaire, showcasing the versatility of Rasch models in various contexts. Considering the specific context of your study, explore how similar techniques or adaptations could be applied to further refine and validate your instrument.

Response 2: Thank you for pointing this out. I agree with this comment. Therefore, I have made an adaptation based on the paper by Cantó-Cerdán et al. (2021) as follows:

1. I Added details in section '2.6. Analysis and Interpretation' are on pages 9 and 10, 292-307.

2. Added section 2.6.1. Rasch Statistics and its explanation on page 10, lines 315-319, and 321-322

3. Explanations about 'Person and Item Reliability' and 'Dimensionality' are divided into different sections on page 10

4. Added section ‘2.6.4. Functionality of the Response Categories on pages 10-11, lines 343-350

5. The explanation of "Differential Item Functioning (DIF)" is made into a section of 2.6.5 to make it easier to understand on page 11, lines 351-358.

6. Added section ‘2.6.6. Targeting' and explanation on page 11, lines 359-365

7. Added section ‘2.6.7. Rasch Rating Scale Method (RSM)' about the frequency of responses Response, Probability graph, and Wright Maps along with explanations on page 11, lines 366-385

8. Add the results of the SILI response based on Rasch RMS on page 19, lines 564-577.

9. I added the Wright maps explanation on lines 620-638.

10. Revise the category probably curve by adding numerical information to the four student answers, namely 1 for the strongly agree option, 2 for the agree option, 3 for the disagree option, and 4 for the strongly disagree option in Figures 15.a, and 15. b. Revisions were also made by adding information under Figure 15. b (in red) on lines 871-873 and 833-885.

Comments 3:

Consideration of Rating Scale Method:

DiStefano and Jiang (2020) [3] discuss the application of the Rasch Rating Scale Method to questionnaire data. Consider incorporating insights from this paper to strengthen the analysis of the attitude features in your study. This could provide additional depth in understanding respondents’ attitudes towards scientific inquiry, leveraging the Rasch Rating Scale Method for more nuanced insights.

Response 3: Thank you for pointing this out. I Agree. I have made the following revisions:

1.      I revised section ‘ 2.5. Data collection by clarifying the use of the Likert scale in the SILI attitude aspect is on page 9, lines 283-286.

2.      I revised the addition of section “ 2.6.7. Rasch Rating Scale Method (RSM) to deepen the discussion of SILI Attitude aspects, found on page 11, lines 366-386

3.      I revised the Probability Curve in Figures 3. a, 3. b, and 3. c by adding a dotted line as a threshold mark between the two answer categories on page 18.

4.      I added an explanation about 'the probability curve Based on Rasch RMS' at SILI response aspects of attitude on page 19, pages 564-577.

5.      I added the discussion of the 'Wright map' into one section, namely the section '3.3. Wright Maps SILI' on page 19 and the explanation on page 22, lines 620-638.

6.      I revised page 30 by adding the RMS model on lines 844-863

Comments 4:

¨      Knowledge Level Calculation in the Rasch Model:

Lutsenko and Seytmanbitov (2022) [4] discuss the calculation of knowledge levels in the Rasch model. Explore the methods and examples presented in this paper to enhance the analysis of knowledge features in your study. This may provide additional insights into how knowledge levels are distributed across different aspects of the scientific inquiry, contributing to a more comprehensive understanding.

Response 4:  Thank you for pointing this out. I can't entirely agree. According to Lutsenko and Seytmanbitov (2022), 'An important result of calculating the level of knowledge in the Rasch model is calculating the test probability of the participant solving all the test questions. These results, together with the a priori distribution of knowledge levels’. Based on Lutsenko and Seytmanbitov's (2022) explanation, this level of knowledge will be helpful in research that focuses on discussing responses from respondents in the form of the probability of test participants completing all test questions. In my study, prospective science students' knowledge level has been explained in detail in the text map regarding the distribution of responses and the difficulty level of items. Then, all participants complete all the test questions, so calculating the 'level of knowledge' is unimportant. This condition is based on my research objectives, which focus on developing SILI and respondents' responses to SILI items.

Comments 5: What is the ‘NoS’ mentioned at line 32 ?  Please clear to a non-expert audience,

Response 5: Thank you for pointing this out. I Agree. I have revised by adding the abbreviation NoS on page 1, lines 30-31. I also added about Nos and researched it on pages 1-2, lines 35-49.

Comments 6: There is one value with comma in Table 5, line 278, last but not least on the right in first row. Please uniform to the other values they are with points,

Response 6:

Thank you for pointing this out. I Agree. I have revised one value, which was initially written with commas, to 1.66 on page 12 and 0.19 on page 13 (uniform to the other values using points)

Comments 7: Please check at line 293 whether in Anglo-Saxon style after semicolon you should start with a small letter instead that with a capital letter (…of 1.69; The value of the SILI person…),

Response 7:

Thank you for pointing this out. I Agree. I have checked line 293 in Anglo-Saxon style after the semicolon starts with a small letter. So … of 1.69; The value of the SILI person… is changed to …… of 1.69; the value of the SILI person……. Found on page 12, lines 409-410.

Comments 8: Please check and correct in third column first row at Table 6 line 321 ‘JMLE masure’,

Response 8: Thank you for pointing this out. I Agree. I corrected the third column, the first row at Table 6, line 321, 'JMLE measure,' and revised it to 'JMLE measure.'

Comments 9: If I were you, I would put paragraph lines 383-390 before Figure 2.a, 2.b, and 2.c, so that I could learn before what the acronyms ICC (Item Characteristic Curve) and IRF (Item Response Function) respectively mean. Think also to renumber figures with letters ‘a’ to ‘c’ as you did in the case of Figure 3.a 3.b, and 3.c,

Response 10:

Thank you for pointing this out. I Agree. I have revised by putting paragraph lines 383-390 before 'Figure 2. a, 2. b, and 2. c' so that I could learn before what the acronyms ICC (Item Characteristic Curve) and IRF (Item Response Function), respectively, mean, found on page 16 lines 502-504. I also revised Figure 2 and renumbered it with letters 'a' to 'c' to become Figures 2. a, 2. b, and 2. c on page 16, lines 499-500, and page 17.

Comments 11: Figure 3.a 3.b, and 3.c are indicated differently at line 408 of the main body (they are without point between number and letter). Please choose one only method to indicate figures along the text,

Response 11: Thank you for pointing this out. I Agree. I have revised it by putting a period between the numbers and letters on page 17, lines 538-540. So it becomes "The probability of response to the SILI knowledge aspect is shown in Figure 3. a. Meanwhile, the probability of response to the SILI skills aspect is shown in Figure 3. b. The probability of response to the SILI attitude aspect is shown in Figure 3. c.”

Comments 12: The acronym of Differential Item Functioning (DIF) is already solved in Table 10 at line 494, so that you do not need to solve it again in Table 11 at line 575, as you already did in the previous Figure 5–6–7,

Response 12:

Thank you for pointing this out. I Agree. I have removed 'The acronym of Differential Item Functioning (DIF)' in Tables 11 and 12. Then, based on suggestions from reviewers, I changed tables 10, 11, and 12 to figures 5, 9, and 10 to make it easier to understand and save place, 'The acronym of Differential Item Functioning (DIF)' is on page 11 lines 351-358.

Comments 13: What is the asterisk at line 600 in Figure 8? Does this indicate that a residual category is to be calculated?

Response 13: Thank you for pointing this out. The asterisk at line 600 in Figure 8 shows the average between the DIF Measures of both genders. Revisions have been made with the addition of an asterisk on page 24 (red text), lines 673-674.

Comments 14: Please mention reference no. 5 in the following way:

Yannier, N., Hudson, S. E., Koedinger, K. R., Hirsh-Pasek, K., Golinkoff, R.M., Munakata, Y. et al. (2021). Active learning: "Hands-on" meets "minds-on". Science. Oct;374(6563):26-30. doi: 10.1126/science.abj9957. Epub 2021 Sep 30. PMID: 34591619.

Response 14: Thank you for pointing this out. I Agree. I have revised it by mentioning reference no. 23 in the following way:

Yannier, N., Hudson, S. E., Koedinger, K. R., Hirsh-Pasek, K., Golinkoff, R. M., Munakata, Y. et al. (2021). Active learning: "Hands-on" meets "minds-on". Science. Oct;374(6563):26-30. doi: 10.1126/science.abj9957. Epub 2021 Sep 30. PMID: 34591619.

This can be seen on page 33, lines 958-959.

Comments 13: Please mention reference no. 23 in the following way:

Brownscombe, J. W., Lédée, E. J. I., Raby, G. D. et al. (2019). Conducting and interpreting fish telemetry studies: considerations for researchers and resource managers. Rev Fish Biol Fisheries 29, 369–400. https://doi.org/10.1007/s11160-019-09560-4,

Response 13: Thank you for pointing this out. I Agree. I have revised it by mentioning reference no. 23 in the following way:

Brownscombe, J. W., Lédée, E. J. I., Raby, G. D. et al. (2019). Conducting and interpreting fish telemetry studies: considerations for researchers and resource managers. Rev Fish Biol Fisheries 29, 369–400. https://doi.org/10.1007/s11160-019-09560-4.

This can be seen on page 34, lines 997-998.

Comments 14: Reference no. 38: the dot after the year of publication between parentheses is missing,

Response 14: Thank you for pointing this out. I Agree. I have revised ‘Reference No. 38' by adding a period after the year of publication.

Odegaard M, Haug B, Mork S, & Sorvik G O. (2015). Budding Science and Literacy. A Classroom Video Study of the Challenges and Support in an Integrated Inquiry and Literacy Teaching Model. Proc. Social and Behavioral Sci. 167, 274 – 278, Can be seen on page 34, lines 1026-1027.

Comments 15: Please mention reference no. 51 in the following way:

Dahlgren A, Semakula D, Chesire F et al. Critical thinking about treatment effects in Eastern Africa: development and Rasch analysis of an assessment tool [version 1; peer review: awaiting peer review]. F1000Research 2023, 12:887. https://doi.org/10.12688/f1000research.132052.1.

Response 15:

Thank you for pointing this out. I Agree. I have revised ‘ reference no. 51' in the following way:

Dahlgren A, Semakula D, Chesire F et al. Critical thinking about treatment effects in Eastern Africa: development and Rasch analysis of an assessment tool [version 1; peer review: awaiting peer review]. F1000Research 2023, 12:887. https://doi.org/10.12688/f1000research.132052.1, can be seen on page 34, lines 1051-1053.

Reviewer 3 Report

Comments and Suggestions for Authors

Dear Authors,

Your paper on the "Development and Validation of Scientific Inquiry Literacy Instrument (SILI) Using the Rasch Measurement Model" is a significant contribution to the field. However, I recommend a few modifications to enhance its clarity and comprehensiveness:

Inclusion of Sample Questions: To aid readers in understanding the scope and depth of SILI, consider including a selection of sample questions in the main body of the paper or as a comprehensive list in an appendix. This will provide a clearer picture of what SILI measures and how it operates.

Additional Research on NOS (Nature of Science) in Science Teachers: Your paper would benefit from incorporating more studies on the Nature of Science (NOS) concerning potential science teachers. This could provide a richer context for the application of SILI and its relevance to teacher education.

Conciseness in Result Presentation: Aim to present the results more succinctly. While detailed data is valuable, a more condensed presentation, perhaps with the aid of clear tables or figures, would enhance readability and allow readers to grasp the key findings more easily.

Elaboration on Research Limitations: Please provide a more detailed discussion of the limitations of your study, particularly regarding the choice of the sample. This transparency is crucial for readers to fully understand the scope and applicability of your findings.

These suggestions aim to augment the impact and clarity of your work. Your contribution to the field is appreciated, and I look forward to the revised version of your paper.

Best regards,

Author Response

Response to Reviewer X Comments

1. Summary

2. Questions for General Evaluation

Reviewer’s Evaluation

Response and Revisions

Does the introduction provide sufficient background and include all relevant references?

Yes/Can be improved/Must be improved/Not applicable

I have revised it by adding some information to the introduction and rearranging the position of paragraphs that are not suitable. My corresponding response is presented in the point-by-point response letter, the same as below.

Are all the cited references relevant to the research?

Yes/Can be improved/Must be improved/Not applicable

I have revised it by adding 15 references relevant to the research. Additional references can be seen in the bibliography.

Is the research design appropriate?

Yes/Can be improved/Must be improved/Not applicable

Revisions are made by explaining the appropriate design in more detail by looking at Figure 1 to explain the appropriate research design.

Are the methods adequately described?

Yes/Can be improved/Must be improved/Not applicable

Revisions have been made to the Sequential Exploratory Research Procedure on the Development of SILI, presented in Figure 1, and the method explanation has been added on Page 4, lines 176-190.

Addition of sections 2.3 and 2.4 to become part of section 2. Materials and methods. Then add information on page 6, lines 230-234 and 242-234.

Added data collection information on page 6, lines 267-271, 283-284,

I added an explanation of analysis and interpretation on pages 9 to 10, lines 292-306. Then Added an explanation section about person and item reliability, dimensionality, Functionality of the Response Categories, Differential Item Functioning (DIF), Targeting, and Rasch Rating Scale Method (RSM) on pages 10-11.

Are the results clearly presented?

Yes/Can be improved/Must be improved/Not applicable

A revision has been carried out to explain the indicators and aspects of the resulting SILI by adding information to clarify the findings.

Revisions are carried out by correcting typos, writing errors, and sentences that could be better structured. It can be seen on page 12, lines 409-410, Table 6, page 16, lines 499-500, and page 17.

I added an explanation of Rasch RMS on page 19, lines 564-577.

I added section 3.3. Wright Maps SILI is on page 19, line 578.

I Improved the display of the wrigh map by adding captions and arrows for easy reading on pages 20-21.

I added the Wright maps explanation on lines 620-638.

I revised section '3.4. Detection of Gender-Biased', Table 10 is simplified into Figure 5 and adds some information in lines 673-674.

I revised section '3.4. Detection of Gender-Biased', Table 10 is simplified into Figure 5 and adds some information in lines 673-674.

I revised section '3.5. For detecting "region-biased"', Table 11 is simplified into Figure 9.

In section 3.5, Table 12 is simplified into Figure 10.

Revisions have been made by adding examples of questions used in appendices 1, 2, and 3.

Revisions were made by adding Rasch RMS information on page 30, lines 844-863.

Revisions were made by adding information about probability curves on page 31, lines 871-873 and 833-886.

Are the conclusions supported by the results?

Yes/Can be improved/Must be improved/Not applicable

Revisions were made to cunculation and limitation by removing repeated sentences in line 895 and improving the conclusion by adding lines 900-903.

3. Point-by-point response to Comments and Suggestions for Authors

Comments 1:

Inclusion of Sample Questions: To aid readers in understanding the scope and depth of SILI, consider including a selection of sample questions in the main body of the paper or as a comprehensive list in an appendix. This will provide a clearer picture of what SILI measures and how it operates.

Response 1: Thank you for pointing this out. I agree with this comment. I have added examples of the SILI instrument for knowledge aspects in appendix 1, skills aspects in appendix 2, and attitude aspects in appendix 3, pages 35-37.

Comments 2:

Additional Research on NOS (Nature of Science) in Science Teachers: Your paper would benefit from incorporating more studies on the Nature of Science (NOS) concerning potential science teachers. This could provide a richer context for the application of SILI and its relevance to teacher education.

Response 2:

Thank you for pointing this out. I Agree. I have revised by adding the abbreviation NoS on page 1, lines 30-31. I also added about Nos and researched it on pages 1-2, lines 35-49.

Comments 3:

Conciseness in Results Presentation: Aim to present the results more successfully. While detailed data is valuable, a more condensed presentation, perhaps with the aid of clear tables or figures, would enhance readability and allow readers to grasp the key findings more easily.

Response 3:

Thank you for pointing this out. I Agree. I have revised this by deleting 'Table 10, Results of Differential Item Functioning (DIF) analysis of SILI items' and presenting the data in Figure 5, page 25, lines 66-68. Then I deleted 'Table 11, Result of Differential Item Functioning (DIF) analysis of SILI items' and presented the data in Figure 9. pages 26-27, lines 729-737. Then

I deleted 'Table 12, Result of Differential Item Functioning (DIF) analysis of SILI items' and presented the data in Figure 10, page 27, lines 742-748.

Comments 4:

Elaboration on Research Limitations: Please provide a more detailed discussion of the limitations of your study, particularly regarding the choice of the sample. This transparency is crucial for readers to fully understand the scope and applicability of your findings.

Response 4:

Thank you for pointing this out. I Agree. I have revised this by providing a more detailed discussion of the limitations of my study, particularly regarding the choice of the sample. Revisions were made in the Limitations and Further Directions section on page 32, lines 927-933.

4. Response to Comments on the Quality of English Language

Point 1:

Response 1:-

5. Additional clarifications

-

Round 2

Reviewer 1 Report

Comments and Suggestions for Authors

I believe that the statistical analysis carried out is comprehensive and provides information in relation to the questions asked. The problem is that looking at the questions I consider that the study is too ambitious and the results do not show what the authors stated at the beginning of the research. If we look for example at the questions related to attitudes, we see that the respondents are asked for their opinion on the subject, which does not reflect the reality of the situation itself. An individual may think that the best attitude is "X" and that does not mean that he or she possesses that attitude or that he or she puts it into practice.

If the design of the questionnaire used, even if validated by "experts", is not valid, the results will not be valid either. I recognise the effort behind this research, but it is so ambitious that in the end nothing comes out of it. I would try to remove the part on attitudes and the rest would be clearer and more objective.

Author Response

For research article

Response to Reviewer X Comments

1. Summary

2. Questions for General Evaluation

Reviewer’s Evaluation

Response and Revisions

Does the introduction provide sufficient background and include all relevant references?

Yes/Can be improved/Must be improved/Not applicable

·         I have revised it by removing the introduction related to the attitude aspect. Introduction SILI aspects of knowledge and skills are maintained.

·         Removing the research question 'What are the SILI indicators in the attitude aspect to evaluate science teachers' SILI?' means that the number of research questions, initially 5, became 4.

Are all the cited references relevant to the research?

Yes/Can be improved/Must be improved/Not applicable

Are all the cited references relevant to the research

Is the research design appropriate?

Yes/Can be improved/Must be improved/Not applicable

·         Revisions are made by deleting the part explaining SILI's attitude aspect at “2. Materials and Methods”

Are the methods adequately described?

Yes/Can be improved/Must be improved/Not applicable

·         Revisions are made by deleting "Table 4. SILI Indicators for Attitude Aspects," and the Explanation paragraph of Table 4 is also deleted.

·         I removed the 'data collection' section, which explains the attitude aspects of SILI.

Are the results clearly presented?

Yes/Can be improved/Must be improved/Not applicable

·      Revisions are made by deleting "Table 4. SILI Indicators for Attitude Aspects," and the Explanation paragraph of Table 4 is also deleted.

·      I removed the 'data collection' section, which explains the attitude aspects of SILI.

·      I have made revisions by removing the 'Functionality of the Response Categories' and 'Rasch Rating Scale Method (RSM)' sections in section 2.6. Analysis and Interpretation.

·      Deleting the Cronbach Alpha, Reliability, MNSQ, and Separation of SILI aspects attitude in Table 5 and their descriptions.

·      Removed "Item Characteristics of the SILI Attitude Aspect" in the '3. results and discussion' section.

·      Deleting Figures 2.a. Suitability level of the Items S8 Figure 2. b. Suitability level of the Items S4 Figure 2. c Level of Suitability of the Items S3 and explanation.

·      Removing the probability of response to the SILI attitude aspect is shown in Figure 3. c.

·      Removed discussion of RMS SILI response aspects of attitude

·      Deleting Figure 4. c shows the Person-item SILI data map for the attitude aspect and its discussion

·      Deleting the SILI attitude Wrigh map and explanation

·      Rearranging the position of the explanation of Wright maps 4. a and 4. b below the image.

·      Deleting the Dimensionality of the SILI Attitude Aspect results in Table 9 and its discussion.

·      Removed SILI attitude aspect in section '3.4. Detection of Gender-Biased'

·      Removing SILI attitudes in Figure 5. Result of Probability Analysis of SILI Items on Gender and its Explanation.

·      Removed 'Person DIF Gender on Attitude aspect' in figure 8 and explanation

·      Removing SILI attitudes in Figure 7. 'Result of Probability Analysis of SILI items on Region' and explanation.

·      Removing SILI attitudes in Figure 9. Result of Probability Analysis of SILI items on Science Majors.

·      Removed follow-up analysis of attitude aspects of SILI

Are the conclusions supported by the results?

Yes/Can be improved/Must be improved/Not applicable

Revisions were carried out by deleting conclusions about the SILI attitude aspect.

3. Point-by-point response to Comments and Suggestions for Authors

Comments 1:

I believe that the statistical analysis carried out is comprehensive and provides information in relation to the questions asked. The problem is that looking at the questions I consider that the study is too ambitious and the results do not show what the authors stated at the beginning of the research. If we look for example at the questions related to attitudes, we see that the respondents are asked for their opinion on the subject, which does not reflect the reality of the situation itself. An individual may think that the best attitude is "X" and that does not mean that he or she possesses that attitude or that he or she puts it into practice.

If the design of the questionnaire used, even if validated by "experts", is not valid, the results will not be valid either. I recognise the effort behind this research, but it is so ambitious that in the end nothing comes out of it. I would try to remove the part on attitudes and the rest would be clearer and more objective.

Response 1: Thank you for pointing this out. I agree with this comment. The attitude aspects of SILI and its discussion have been removed in every part of the manuscript, from the introduction to the appendices, with the following details:

·         I have revised it by removing the introduction related to the attitude aspect. Introduction SILI aspects of knowledge and skills are maintained on page 3 line 147-148, 187-188.

·         Removing the research question 'What are the SILI indicators in the attitude aspect to evaluate science teachers' SILI?' means that the number of research questions, initially 5, became 4 on page 3-4 line 151-156.

·         Revisions are made by deleting the part explaining SILI's attitude aspect at “2. Materials and Methods” on page 4 line 159-160.

·         I removed the SILI's attitude aspect pada Figure 1.

·         Eliminating the SILI attitude aspect of the SILI on page 6 line 231-234, on page 10 line 356-357, and page 17 line 500-501.

·         Revisions are made by deleting "Table 4. SILI Indicators for Attitude Aspects," on page 10 line 341-342, and the Explanation paragraph of Table 4 is also deleted.

·         I removed the 'data collection' section, which explains the attitude aspects of SILI.

·         I have made revisions by removing the 'Functionality of the Response Categories' and 'Rasch Rating Scale Method (RSM)' sections in section 2.6. Analysis and Interpretation.

·         Deleting the Cronbach Alpha, Reliability, MNSQ, and Separation of SILI aspects attitude in Table 5 and their descriptions.

·         Removed "Item Characteristics of the SILI Attitude Aspect" in the '3. results and discussion' section.

·         Deleting Figures 2.a. Suitability level of the Items S8 Figure 2. b. Suitability level of the Items S4 Figure 2. c Level of Suitability of the Items S3 and explanation.

·         Removing the probability of response to the SILI attitude aspect is shown in Figure 3. c.

·         Removed discussion of RMS SILI response aspects of attitude

·         Deleting Figure 4. c shows the Person-item SILI data map for the attitude aspect and its discussion

·         Deleting the SILI attitude Wrigh map and explanation

·         Rearranging the position of the explanation of Wright maps 4. a and 4. b below the image.

·         Deleting the Dimensionality of the SILI Attitude Aspect results in Table 9 and its discussion.

·         Removed SILI attitude aspect in section '3.4. Detection of Gender-Biased'

·         Removing SILI attitudes in Figure 5. Result of Probability Analysis of SILI Items on Gender and its Explanation.

·         Removed 'Person DIF Gender on Attitude aspect' in figure 8 and explanation.

·         Removing SILI attitudes in Figure 7. 'Result of Probability Analysis of SILI items on Region' and explanation.

·         Removing SILI attitudes in Figure 9. Result of Probability Analysis of SILI items on Science Majors.

·         Removed follow-up analysis of attitude aspects of SILI.

·         Revisions were carried out by deleting conclusions about the SILI attitude aspect on page 22 line 633-635.

·         Removed the attachment to the attitude aspect instrument in SILI

4. Response to Comments on the Quality of English Language

Point 1:

-

5. Additional clarifications

-